# Analysis of meiosis in *Pristionchus pacificus* reveals plasticity in homolog pairing and synapsis in the nematode lineage

Regina Rillo-Bohn[1,2†], Renzo Adilardi[1,2†], Therese Mitros[1], Barış Avşaroğlu[1,2], Lewis Stevens[1,3], Simone Köhler[1,2‡], Joshua Bayes[1], Clara Wang[1,2], Sabrina Lin[1,2], K Alienor Baskevitch[1,2], Daniel S Rokhsar[1,4,5,6], Abby F Dernburg[1,2,7,8]*

[1]Department of Molecular and Cell Biology, University of California, Berkeley, Berkeley, United States; [2]Howard Hughes Medical Institute, Chevy Chase, United States; [3]Darwin Tree of Life Project, Wellcome Sanger Institute, Cambridge, United Kingdom; [4]Department of Energy Joint Genome Institute, Berkeley, United States; [5]Okinawa Institute of Science and Technology Graduate University, Onna, Japan; [6]Chan Zuckerberg Biohub, San Francisco, United States; [7]Biological Systems and Engineering Division, Lawrence Berkeley National Laboratory, Berkeley, United States; [8]California Institute for Quantitative Biosciences, Berkeley, United States

*For correspondence:
afdernburg@berkeley.edu

[†]These authors contributed equally to this work

Present address: [‡]Cell Biology and Biophysics Unit, European Molecular Biology Laboratory, Heidelberg, Germany

Competing interest: The authors declare that no competing interests exist.

**Abstract** Meiosis is conserved across eukaryotes yet varies in the details of its execution. Here we describe a new comparative model system for molecular analysis of meiosis, the nematode *Pristionchus pacificus*, a distant relative of the widely studied model organism *Caenorhabditis elegans*. *P. pacificus* shares many anatomical and other features that facilitate analysis of meiosis in *C. elegans*. However, while *C. elegans* has lost the meiosis-specific recombinase Dmc1 and evolved a recombination-independent mechanism to synapse its chromosomes, *P. pacificus* expresses both DMC-1 and RAD-51. We find that SPO-11 and DMC-1 are required for stable homolog pairing, synapsis, and crossover formation, while RAD-51 is dispensable for these key meiotic processes. RAD-51 and DMC-1 localize sequentially to chromosomes during meiotic prophase and show nonoverlapping functions. We also present a new genetic map for *P. pacificus* that reveals a crossover landscape very similar to that of *C. elegans*, despite marked divergence in the regulation of synapsis and crossing-over between these lineages.

## Introduction

All sexually reproducing organisms rely on the specialized cell division process of meiosis to generate haploid gametes from diploid precursors. Upon fertilization, gametes fuse and restore the diploid chromosome complement in the zygote. This sexual cycle of meiosis and fertilization was likely present in the last eukaryotic common ancestor (*Goodenough and Heitman, 2014*). Studies in plants, animals, fungi, and protists have revealed intriguing diversity in the details of meiotic mechanisms.

A defining feature of meiosis is a 'reductional' division in which homologous chromosomes are separated, usually during the first of two nuclear divisions. A highly choreographed series of chromosome transactions precedes this division and ensures faithful homolog segregation: (1) pairing, in which chromosomes contact and recognize their homologous partners; (2) synapsis, defined as the assembly of a protein ensemble called the synaptonemal complex (SC) between homologs, which stabilizes pairing and leads to their lengthwise alignment; and (3) crossover (CO) recombination, which

creates physical links between chromosomes that promote proper biorientation during anaphase I. Failure to form at least one 'obligate' CO between each pair of homologs typically results in missegregation and aneuploid gametes (*Zickler and Kleckner, 2015*).

Although homolog pairing, synapsis, and CO recombination during meiosis are nearly ubiquitous among eukaryotes, key aspects of these chromosomal transactions show remarkable diversity across different lineages. In most model fungi, plants, and animals, synapsis is thought to be triggered by early steps in recombination. Meiotic recombination is initiated by double-strand breaks (DSBs) catalyzed by the conserved topoisomerase-like enzyme Spo11 (*Bergerat et al., 1997*; *Keeney et al., 1997*). DSBs are resected to form 3' single-stranded DNA overhangs. Dmc1, a meiosis-specific recombinase, forms filaments along the resulting ssDNA segments and promotes interhomolog strand invasion to form joint molecules (JMs). In some organisms, Rad51 is required as a cofactor for Dmc1. A subset of these JMs is ultimately repaired through a meiosis-specific pathway that generates COs, while the rest are mostly repaired through a more 'mitotic-like' homologous recombination pathway to yield non-CO products (*Hunter, 2015*; *Kohl and Sekelsky, 2013*).

Spo11-dependent induction of DSBs and Dmc1-dependent strand invasion are crucial for synapsis, and thus for stable homolog pairing, in the budding yeast *Saccharomyces cerevisiae*, the flowering plant *Arabidopsis thaliana,* and in mice (*Bishop et al., 1992*; *Couteau et al., 1999*; *Grelon et al., 2001*; *Pittman et al., 1998*; *Rockmill et al., 1995*; *Yoshida et al., 1998*). Thus, the formation of JMs is considered to be an intrinsic part of the pairing process by which chromosomes recognize their partners. In contrast, recombination-independent mechanisms of pairing and synapsis have been characterized in other prominent model systems, including the dipteran insect *Drosophila melanogaster* and the nematode *Caenorhabditis elegans* (*Lake and Hawley, 2012*; *Rog and Dernburg, 2013*). While crossing-over is essential for successful execution of meiosis in *C. elegans* and in female fruit flies, homolog pairing and synapsis can be uncoupled from DSB induction and JM formation. In *D. melanogaster* females, pairing initiates in proliferating germline cells even before they enter meiosis and is stabilized by SC formation during early prophase (*Christophorou et al., 2015*; *Christophorou et al., 2013*). *D. melanogaster* males lack both recombination and SCs and have apparently evolved a distinct mechanism to stabilize homolog pairing and enable reductional segregation (*McKee et al., 2012*). In *C. elegans*, pairing and synapsis are both mediated by pairing centers (PCs), specialized sites on each chromosome bound by a family of zinc-finger proteins that interact with nuclear envelope proteins and promote large-scale chromosome movement (*MacQueen et al., 2005*; *Phillips et al., 2005*; *Phillips and Dernburg, 2006*; *Sato et al., 2009*). While interactions between chromosomes and nuclear envelope proteins play important roles in meiotic pairing and synapsis across eukaryotes, in *C. elegans* they have acquired a critical role in coupling homolog pairing to synapsis initiation (*Penkner et al., 2007*; *Penkner et al., 2009*; *Sato et al., 2009*).

To investigate how the meiotic program is modified during evolution, we have established tools to investigate meiosis in the free-living nematode *Pristionchus pacificus,* which has been established as a model for studies in development, evolution, and ecology (*Sommer, 2015*). Like its distant relative *C. elegans*, *P. pacificus* is an androdioecious species, with populations consisting of self-fertilizing hermaphrodites (XX) and a low frequency of spontaneous males (XO) (*Sommer, 2015*; *Sommer et al., 1996*). *P. pacificus* also shares with *C. elegans* a short life cycle of 3.5 days at 20 °C, produces large broods of about 200 progeny by self-fertilization, and is easily cultured in the lab (*Hong and Sommer, 2006*). Although *C. elegans* and *P. pacificus* diverged an estimated 60–90 million years ago (*Werner et al., 2018*), they share the same number of chromosomes (2n = 12). Other than one major chromosomal translocation, macrosynteny is largely maintained between the two species (*Dieterich et al., 2008*; *Rödelsperger et al., 2017*) as well as other Rhabditids. Like other nematodes, *P. pacificus* is thought to have holocentric chromosomes, which we confirm here through cytological analysis. Recent improvements in the genome assembly (*Rödelsperger et al., 2017*) and advances in genome editing (*Lo et al., 2013*; *Namai and Sugimoto, 2018*; *Witte et al., 2015*) have facilitated investigation of developmental and cell biological processes at a more mechanistic level.

We became interested in exploring meiosis in *P. pacificus* in part because of differences in the inventory of meiotic genes predicted from its genome sequence compared to *C. elegans*. In particular, genome sequencing and annotation have revealed the presence of genes encoding an ortholog of Dmc1 as well as two Dmc1 cofactors, Mnd1 and Hop2, all of which are absent from the entire *Caenorhabditis* clade (*Dieterich et al., 2008*). Loss of Dmc1 correlates with the adaptation of

recombination-independent mechanisms for pairing and synapsis in *Drosophila* and *Caenorhabditis* (*Villeneuve and Hillers, 2001*). Therefore, we were keen to examine how homologous chromosomes pair and synapse in *P. pacificus*. In addition, genetic linkage maps have suggested that multiple COs typically occur per chromosome pair during meiosis in *P. pacificus* (*Hong and Sommer, 2006*; *Srinivasan et al., 2003*; *Srinivasan et al., 2002*), while only a single CO per chromosome pair normally occurs during meiosis in *C. elegans*, suggesting potential differences in CO regulation.

Using genome editing, cytogenetic analysis, and recombination mapping, we have characterized the early events of meiotic prophase in *P. pacificus*. We show that homolog pairing, synapsis, and CO recombination are dependent on *Ppa-spo-11* and *Ppa-dmc-1*, while *Ppa-rad-51* is not essential for meiosis. We find that CO sites are designated very early during meiotic prophase, prior to completion of synapsis, and are limited to one per chromosome pair. We also present a new genetic map, which corroborates our cytological evidence that a single CO normally occurs between each pair of homologs per meiosis in both spermatocytes and oocytes. The map also provides evidence of CO suppression and segregation distortion of specific chromosomes in hybrids, indicative of reproductive barriers between strains. Our work highlights both conserved and flexible features of the meiotic program within the nematode lineage and establishes a platform for future investigation.

## Results

### *P. pacificus* as a comparative model system for meiosis

The morphology and organization of the *P. pacificus* germline are very similar to that of *C. elegans*. Hermaphrodites have two tubelike gonad arms in which sperm and ova are produced sequentially, while males have a single arm (*Rudel et al., 2005*). These are organized as a cylindrical monolayer of cells abutting a central core, or rachis. The distal tip is populated by proliferating germline stem cells (*Figure 1A, B and D*), as confirmed by the incorporation of microinjected fluorescently labeled nucleotides into replicating DNA (*Figure 1B*). The injected nucleotides label nuclei at various stages of S-phase; some preferentially incorporate fluorescence into the X-chromosomes, which are silenced in the germline, as in *C. elegans* (*Kelly et al., 2002*). The onset of meiotic prophase is marked by an obvious change in nuclear organization in which fluorescently stained chromosomes form a conspicuous crescent-shaped mass (*Figure 1A*). Immediately proximal to this 'transition zone,' DAPI staining reveals parallel tracks, indicative of paired and synapsed homologous chromosomes at the pachytene stage. Gametogenesis switches from spermatogenesis to oogenesis during early adulthood. As oocytes approach maturation, chromosomes undergo striking decondensation between diplotene and diakinesis, a stage that has been referred to as the 'growth zone' (*Rudel et al., 2005*). Similar chromosome morphology is observed during a late 'diffuse stage' of meiotic prophase in many other eukaryotes (*Zickler and Kleckner, 1999*), but is rarely seen in *C. elegans*. We use the term 'diffuse stage' for consistency with other organisms. In the most proximal region of the hermaphrodite gonad, oocytes mature and form a single row of large cells, and chromosomes condense dramatically as the nuclei grow in size. Six bivalents can be detected as compact DAPI-staining bodies in oocytes at diakinesis, the last stage of meiotic prophase preceding the first meiotic division (*Figure 1A and C*, *Rudel et al., 2005*; *Sommer et al., 1996*).

It has been assumed that chromosomes in *P. pacificus* are holocentric, as in *C. elegans*, but we are unaware of direct evidence to support this idea. We thus identified a gene encoding CENP-C (HCP-4 in *C. elegans*), a conserved kinetochore protein, in the *P. pacificus* genome and inserted a V5 epitope tag at its 3' end. The distribution of CENP-C on mitotic chromosomes in embryos and mitotic germ cells confirmed their holocentric organization (*Figure 1D*). Kinetochores appeared as linear structures along the full length of each chromatid on mitotic chromosomes, rather than discrete foci. CENP-C also coated the chromosomes during meiotic metaphase I (*Figure 1C*), as in *C. elegans* (*Shakes et al., 2009*).

### Stable homolog pairing requires early recombination factors

BLAST searches of the *P. pacificus* genome revealed an open-reading frame encoding an unambiguous ortholog of Dmc1, a meiosis-specific paralog of Rad51 (*Supplementary file 1*). Orthologs of the Dmc1 cofactors Mnd1 and Hop2 were also identified by homology searches (*Figure 1—figure supplements 3–5*; *Figure 2—figure supplement 2*). By contrast, Dmc1/Mnd1/Hop2 are absent from both *C.*

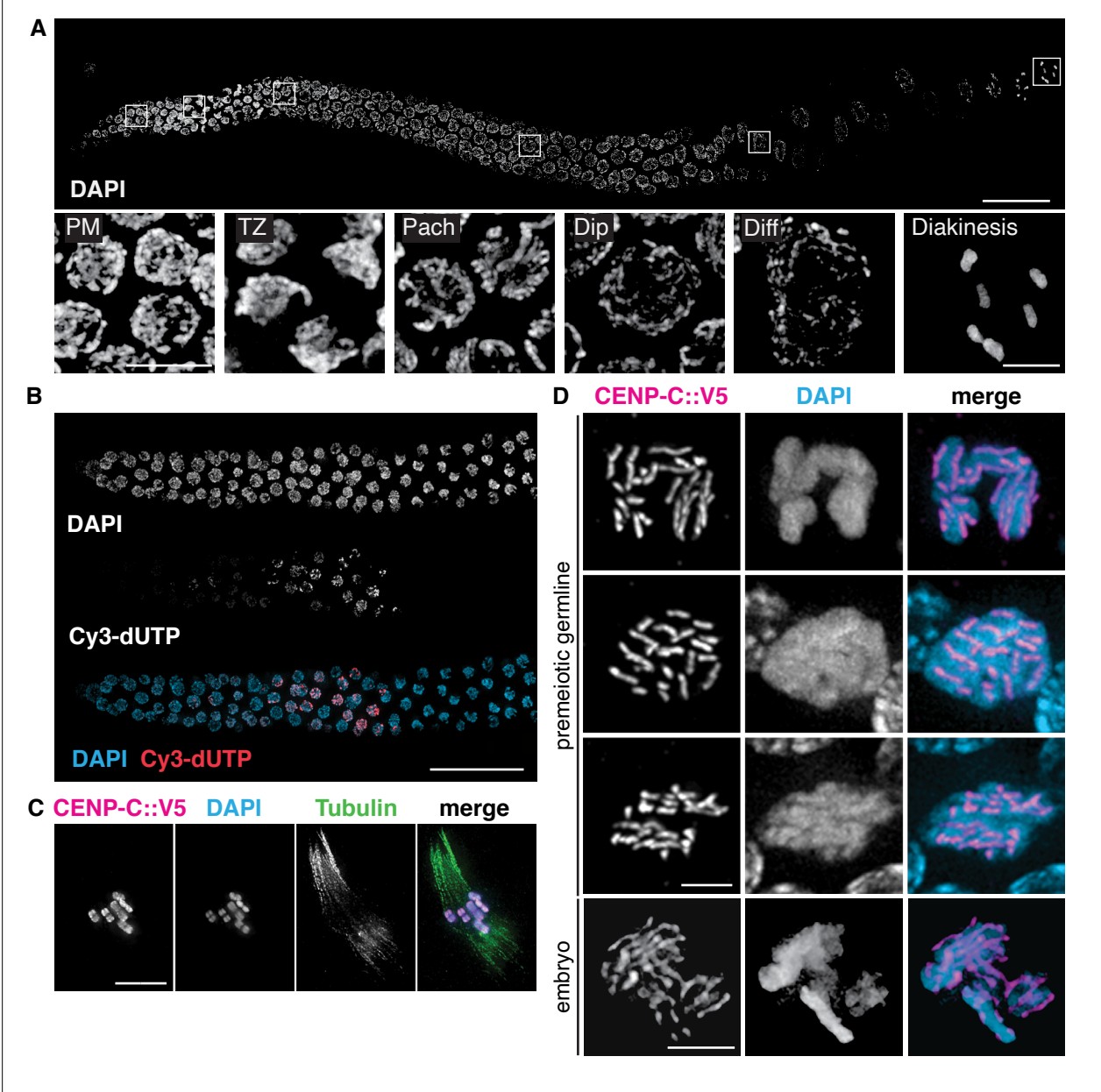

**Figure 1.** Germline organization and meiotic nuclear morphology in *P. pacificus* are superficially similar to *C. elegans*. (**A**) Projection image of the distal arm of a *P. pacificus* hermaphrodite gonad stained with DAPI. Scale bar, 30 μm. Insets show representative nuclei from the premeiotic region (PM), transition zone (TZ), pachytene (Pach), diplotene (Dip), diffuse stage (Diff), and diakinesis. Scale bar, 5 μm. (**B**) Distal region of a *P. pacificus* germline following injection of fluorescent nucleotides to label replicating DNA. Scale bar, 30 μm. (**C**) Metaphase I oocyte expressing CENP-C::V5, stained with anti-V5, DAPI, and anti-tubulin. Scale bar, 10 μm. (**D**) Mitotic chromosomes (DAPI) in the premeiotic germline of adult hermaphrodites and a 2–4 cell stage embryo and expressing CENP-C::V5 (magenta). Scale bar, 2 μm.

The online version of this article includes the following source data and figure supplement(s) for figure 1:

**Figure supplement 1.** Multiple sequence alignments of homologs of proteins analyzed in this study.

**Figure supplement 1—source data 1.** Amino acid sequences of homologs of SPO-11.

**Figure supplement 2.** Multiple sequence alignment of Rad51/RAD-51 proteins.

**Figure supplement 2—source data 1.** Amino acid sequences of homologs of RAD-51.

**Figure supplement 3.** Multiple sequence alignment of Rad51/RAD-51 proteins.

**Figure supplement 3—source data 1.** Amino acid sequences of homologs of DMC-1.

*Figure 1 continued on next page*

*Figure 1 continued*

**Figure supplement 4.** Multiple sequence alignment of Mnd1/MND-1 proteins.

**Figure supplement 4—source data 1.** Amino acid sequences of homologs of MND-1.

**Figure supplement 5.** Multiple sequence alignment of Hop2/HOP-2 proteins.

**Figure supplement 5—source data 1.** Amino acid sequences of homologs of HOP-2.

**Figure supplement 6.** Multiple sequence alignment of meiotic HORMA domain proteins homologous to *P. pacificus* HOP-1.

**Figure supplement 6—source data 1.** Amino acid sequences of homologs of HOP-1.

**Figure supplement 7.** Multiple sequence alignment of SYP-4 homologs from several nematode species.

**Figure supplement 7—source data 1.** Amino acid sequences of homologs of SYP-4.

**Figure supplement 8.** Multiple sequence alignment of COSA-1/Cntd1 proteins from metazoans.

**Figure supplement 8—source data 1.** Amino acid sequences of homologs of COSA-1.

---

*elegans* and *D. melanogaster*, two model organisms that have evolved recombination-independent mechanisms of homolog pairing and synapsis (*Villeneuve and Hillers, 2001*). We analyzed the genome sequences of other nematodes to determine the evolutionary history of these genes within the nematode lineage. This analysis revealed that Dmc1/Mnd1/Hop2 have been lost several times during the evolution of nematodes, including the entire *Caenorhabditis* genus and all sequenced members of Clade IV (*Figure 2—figure supplement 1*). As expected in light of its essential function in DNA repair, the recombinase Rad51 was detected in all genomes examined (data not shown).

In *C. elegans*, homolog pairing and synapsis require a family of zinc-finger proteins (HIM-8, ZIM-1, ZIM-2, and ZIM-3) that bind to motifs enriched within PC regions near one end of each chromosome (*Phillips et al., 2009b*). We identified homologs of these proteins in many of the genome sequences from Clade V nematodes, but not in *Pristionchus*.

We tested whether homolog pairing in *P. pacificus* depends on DSBs or strand-exchange proteins. To generate *spo-11*, *dmc-1*, and *rad-51* null mutants, we first employed TALEN-mediated gene disruption, and later CRISPR/Cas9 genome editing techniques (this study; *Lo et al., 2013*; *Witte et al., 2015*). Genome editing has thus far been less efficient in *P. pacificus* than in *C. elegans*, and techniques such as co-CRISPR (in which another locus is simultaneously edited to an allele with an obvious visible phenotype) have not been helpful to enrich for the desired edited progeny in our hands or others' (*Witte et al., 2015*; data not shown). Nevertheless, we were readily able to isolate mutant alleles by screening a large number of F1 progeny from injected hermaphrodites (see Materials and methods). Independent alleles isolated from either TALEN- or CRISPR-mediated genome editing resulted in identical mutant phenotypes. All data presented here were based on alleles generated by CRISPR/Cas9. Because balancer chromosomes are not currently available for *P. pacificus*, most mutations described here were maintained in unbalanced heterozygotes, with PCR-based genotyping performed every few generations and immediately before each experiment. Self-fertilization of heterozygotes results in broods with 25% homozygous mutant animals, which can be identified based on their small brood size and high incidence of male self-progeny. Immunostaining was also used to determine the presence or absence of the protein of interest. We also found that a freezing method developed for *C. elegans* that uses DMSO and trehalose as cryoprotectants (Kevin F. O'Connell, Worm Breeders' Gazette, pers. comm.) allowed robust recovery of frozen *P. pacificus*, facilitating strain maintenance and archival storage.

As expected, disruption of either *spo-11* or *dmc-1* resulted in the detection of 12 DAPI-staining univalent chromosomes at diakinesis, indicative of a failure in CO recombination (Figure 5). Surprisingly, *rad-51* mutants were fertile and displayed only mild meiotic defects (see below). We thus validated the loss of RAD-51 function in mutant animals by generating multiple alleles, which showed indistinguishable phenotypes. We also confirmed the absence of RAD-51 protein by immunofluorescence with a polyclonal antibody raised against recombinant Ppa-RAD-51 (*Figure 4—figure supplement 1B*, see Materials and methods). Mutations in *spo-11* or *dmc-1*, but not *rad-51*, resulted in an obvious extension of the region of the germline displaying the crescent-shaped nuclear morphology characteristic of early meiosis (*Figure 2—figure supplement 2*). A similar 'extended transition zone' phenotype is seen in *C. elegans* mutants that fail to synapse their chromosomes during meiosis, suggesting that *spo-11* and *dmc-1* might be required for synapsis in *P. pacificus*.

To visualize and quantify homolog pairing, we generated FISH probes against two short tandem repeats found on chromosomes X and IV (*Figure 2A*). We measured the distance between pairs of homologous FISH signals in individual nuclei for each genotype. To analyze pairing kinetics, we divided the distal gonads into five zones of equal length. In zone 1 in wild-type *P. pacificus* hermaphrodites, which contains mostly proliferating germ cells, pairs of FISH signals remained far apart, with an average distance of 2.4 ± 1.0 µm (SD) and 2.5 ± 0.8 µm for chromosomes X and IV, respectively (*Figure 2B and D*). In zone 2, which spans the transition zone, the average distances between homologous loci decreased significantly (1.2 ± 1.1 µm and 1.1 ± 1.1 µm for probes on chromosomes X and IV, respectively). Surprisingly, homologous FISH probes were more frequently separated in zones 4 and 5 (*Figure 2B and D*). This differs from what is seen in *C. elegans*, where homologous loci remain closely apposed throughout an extended pachytene stage spanning most of the distal region (before the 'loop') of the gonad (*MacQueen, 2002*). Together with our analysis of synapsis (below), this indicated that desynapsis initiates soon after completion of synapsis in *P. pacificus*, resulting in partial separation of homologs.

We noted that the average distances between pairs of homologous FISH signals in *spo-11* and *dmc-1* mutants also decreased markedly upon meiotic entry, although clearly less so than in wild type (*Figure 2D*). In contrast, *rad-51* mutants showed distributions of probe distances more similar to wild-type animals (*Figure 2C and D*). We considered the possibility that the proximity between FISH signals might reflect the clustering of all chromosomes during leptotene/zygotene, rather than specific homologous interactions. If so, the extended transition zone morphology in *spo-11* and *dmc-1* might obscure a pairing defect that would be more apparent in the absence of clustering (*Figure 2—figure supplement 2*). To address this, we measured the distances between pairs of heterologous FISH signals in the premeiotic region (dispersed) versus the transition zone (clustered). We observed that FISH signals on different chromosomes were also significantly closer to each other in the transition zone compared to premeiotic nuclei in both wild-type and mutant animals (*Figure 2E*). The distances between heterologous versus homologous pairs of FISH loci were not significantly different in *spo-11* and *dmc-1* mutants (p=0.1777 and p=0.6774, respectively, by Student's *t*-test), but homologous signals were clearly closer than heterologous signals in wild-type and *rad-51* mutant animals (p<0.0001 by Student's *t*-test; *Figure 2E*). These data support the idea that clustering, rather than specific pairing, promotes proximity between both homologous and heterologous loci during leptotene/zygotene in *spo-11* and *dmc-1* mutants. Although we cannot conclude that transient homologous pairing is absent in these mutants, it is evident that these early recombination factors are required for stable pairing and extended association of homologous loci throughout prophase. In contrast, *rad-51* is dispensable for homolog pairing, as in *C. elegans*.

## SPO-11 and DMC-1 are required for homologous synapsis

To further investigate meiotic progression in *P. pacificus* and to probe the role of early recombination factors in synapsis, we developed cytological markers for the chromosome axis, which normally assembles upon meiotic entry, and the SC, which assembles between paired axes during early prophase. Identification of a candidate axial element component was straightforward due to the presence of the easily recognized HORMA (Hop1, Rev7, Mad2) domain among members of this family of proteins (*Aravind and Koonin, 1998*; *Vader and Musacchio, 2014*). We identified a gene encoding a HORMA domain protein that is most closely related to *C. elegans* HIM-3 by reciprocal BLAST analysis (*Supplementary file 1*; *Figure 1—figure supplement 6*). We refer to this protein as Ppa-HOP-1, after the founding member of the meiotic HORMA proteins, *S. cerevisiae* Hop1 (*Hollingsworth and Byers, 1989*). Using genetic immunization, we raised a polyclonal antibody against a 100-amino acid segment of the predicted protein including part of the HORMA domain and found that it indeed recognized chromosome axes from meiotic entry through late prophase (*Figure 3*).

To enable cytological detection of SC assembly, we searched for homologs of SC proteins. This was not straightforward due to rapid divergence of these proteins in nematodes as well as their extensive regions of coiled-coil potential, which constrains their amino acid composition. *C. elegans* expresses six known SC proteins, SYP-1–6. SYP-1–4 are all required for assembly of the SC, while SYP-5 and SYP-6 are partially redundant (*Hurlock et al., 2020*; *Rog and Dernburg, 2013*). SYP-4 has a distinctive C-terminal domain containing several unusual motifs enriched in glycine and phenylalanine residues, which enabled us to identify it with confidence among the predicted proteins in *P.*

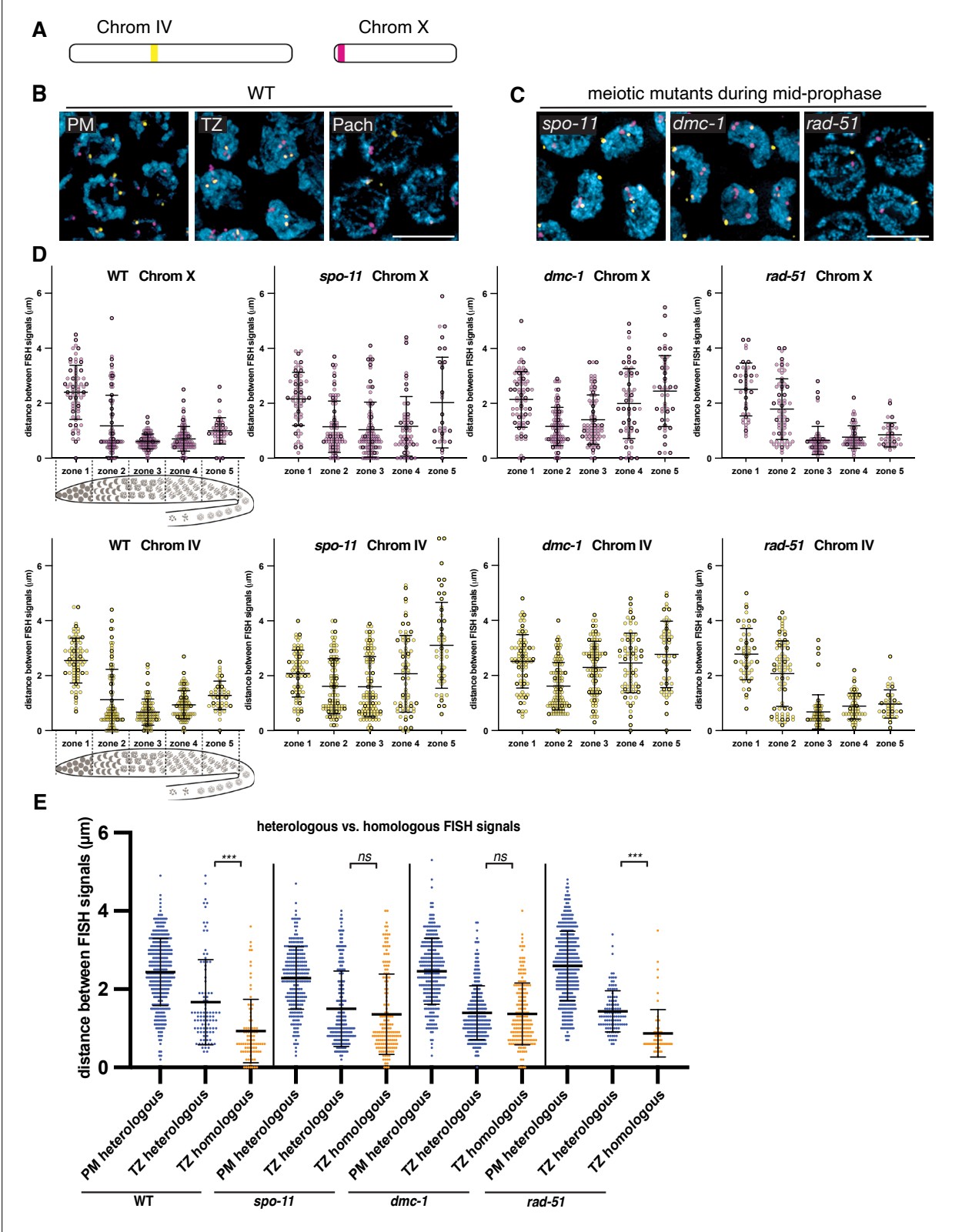

**Figure 2.** Stable homolog pairing requires double-strand breaks (DSBs) and strand invasion. (**A**) Diagram showing the locations of tandem repeat sequences used to generate DNA FISH probes for pairing analysis in *P. pacificus*. (**B**) Representative images show the progression of homolog pairing of chromosome X (magenta) and chromosome IV (yellow) during meiotic prophase in wild-type hermaphrodites. Premeiotic region (PM), transition zone (TZ), and pachytene (Pach). Scale bar, 5 μm. (**C**) Representative images of FISH probe signals in *spo-11*, *dmc-1*, and *rad-51* mutants during mid-prophase

*Figure 2 continued on next page*

*Figure 2 continued*

stage (roughly equivalent to the pachytene stage in wild-type germlines). Scale bar, 5 µm. (**D**) Temporal progression of X and IV chromosome pairing in WT, *spo-11*, *dmc-1*, and *rad-51* mutants. Graphs show the distribution of distances within each of five equally sized zones spanning meiotic prophase. (**E**) Distance between pairs of heterologous FISH signals was measured in premeiotic (PM) and transition zone (TZ) nuclei in WT, *spo-11*, *dmc-1*, and *rad-51* mutants (spanning zones 1 and 2 only). Distances between pairs of homologous FISH signals (Chr. X and IV combined) in TZ nuclei are included for comparison. ***p<0.0001, by Student's *t*-test. See also *Figure 2—figure supplement 1*, *Figure 2—figure supplement 2*, and *Figure 2—source data 1*.

The online version of this article includes the following source data and figure supplement(s) for figure 2:

**Source data 1.** Distances between FISH signals plotted in *Figure 2*.

**Figure supplement 1.** Presence of meiotic pairing proteins across the nematode phylogeny.

**Figure supplement 2.** A prolonged region of polarized nuclear morphology ("transition zone" nuclei) is seen in *spo-11* and *dmc-1*, but not in *rad-51* mutants.

*pacificus* (*Figure 1—figure supplement 2*). We inserted an HA epitope tag at the C-terminus of the endogenous coding sequence and found that immunofluorescence with an epitope-specific antibody localized specifically between paired meiotic chromosomes, confirming SYP-4::HA as a useful marker for the SC (*Figure 3A and B*). A likely *P. pacificus* ortholog of *C. elegans* SYP-1 has also recently been identified, and an epitope-tagged allele displays an identical dynamic distribution (*Kursel et al., 2021*). The tagged SYP-4::HA protein supports normal meiosis, as indicated by the low percentage of inviable embryos and males among the progeny of homozygotes (*Figure 3—source data 1*).

HOP-1 was detected in the nucleoplasm in the premeiotic region of the germline and formed linear structures along chromosomes upon meiotic entry. SYP-4 localized along chromosome segments shortly thereafter, and fully colocalized with HOP-1 tracks during pachytene. Notably, the region of the germline containing nuclei with fully aligned stretches of SYP-4 and HOP-1 was very short compared to *C. elegans*, in which SC disassembly occurs close to the bend of the gonad arm, shortly prior to diakinesis. In contrast, SC disassembly initiated much earlier in *P. pacificus*; the major fraction of prophase nuclei are thus in diplotene since chromosomes are only partially synapsed. Six short stretches of SYP-4 were apparent in these nuclei, which persisted over an extended region (*Figure 3A and B*). This asymmetrical pattern of SC disassembly is highly reminiscent of a more transient diplotene stage in *C. elegans*, in which the SC remains associated with one 'arm' of each homolog pair and subsequently contributes to the stepwise loss of cohesion during the meiotic divisions (*Martinez-Perez et al., 2008*).

HOP-1 localized normally to chromosome axes in *spo-11* and *dmc-1* mutants, but extensive SC assembly failed. Instead, small, dispersed puncta of SYP-4 were observed along chromosome axes, with occasional longer tracks (*Figure 3C*). The number of these tracks was variable. They did not appear to associate preferentially with specific chromosomes, and our pairing analysis indicates that they represent nonhomologous synapsis. In contrast to *spo-11* and *dmc-1* mutants, *rad-51* mutants displayed robust synapsis with a distribution of stages similar to that seen in wild-type hermaphrodites (*Figure 3C*). These observations indicate that homologous synapsis depends on SPO-11 and DMC-1 in *P. pacificus*, in contrast to the recombination-independent synapsis seen in *C. elegans*.

## DMC-1 and RAD-51 localize sequentially during distinct stages of meiotic prophase

To investigate the functions of and interplay between DMC-1 and RAD-51 in *P. pacificus*, we inserted a V5 epitope tag at the C-terminus of DMC-1 using CRISPR/Cas9 and raised a polyclonal antibody that recognizes RAD-51 (see Materials and methods). DMC-1::V5 supported normal meiosis, as evidenced by a normal brood size, high embryonic viability, and low percentage of males (*Figure 3—source data 1*). Surprisingly, the two proteins showed distinct and nonoverlapping patterns of localization. DMC-1 localized very broadly along chromosomes in transition zone nuclei and disappeared immediately upon completion of synapsis. RAD-51 displayed a much more restricted, punctate distribution along chromosomes, which was only detected in nuclei in which DMC-1 no longer coated the chromosomes (*Figure 4A and B*). Occasional nuclei at the border between the transition zone and pachytene region exhibited both DMC-1 and RAD-51, although DMC-1 was very faint in these nuclei and did not overlap with RAD-51 (*Figure 4C*). Additionally, DMC-1 remained strongly associated with chromosomes in

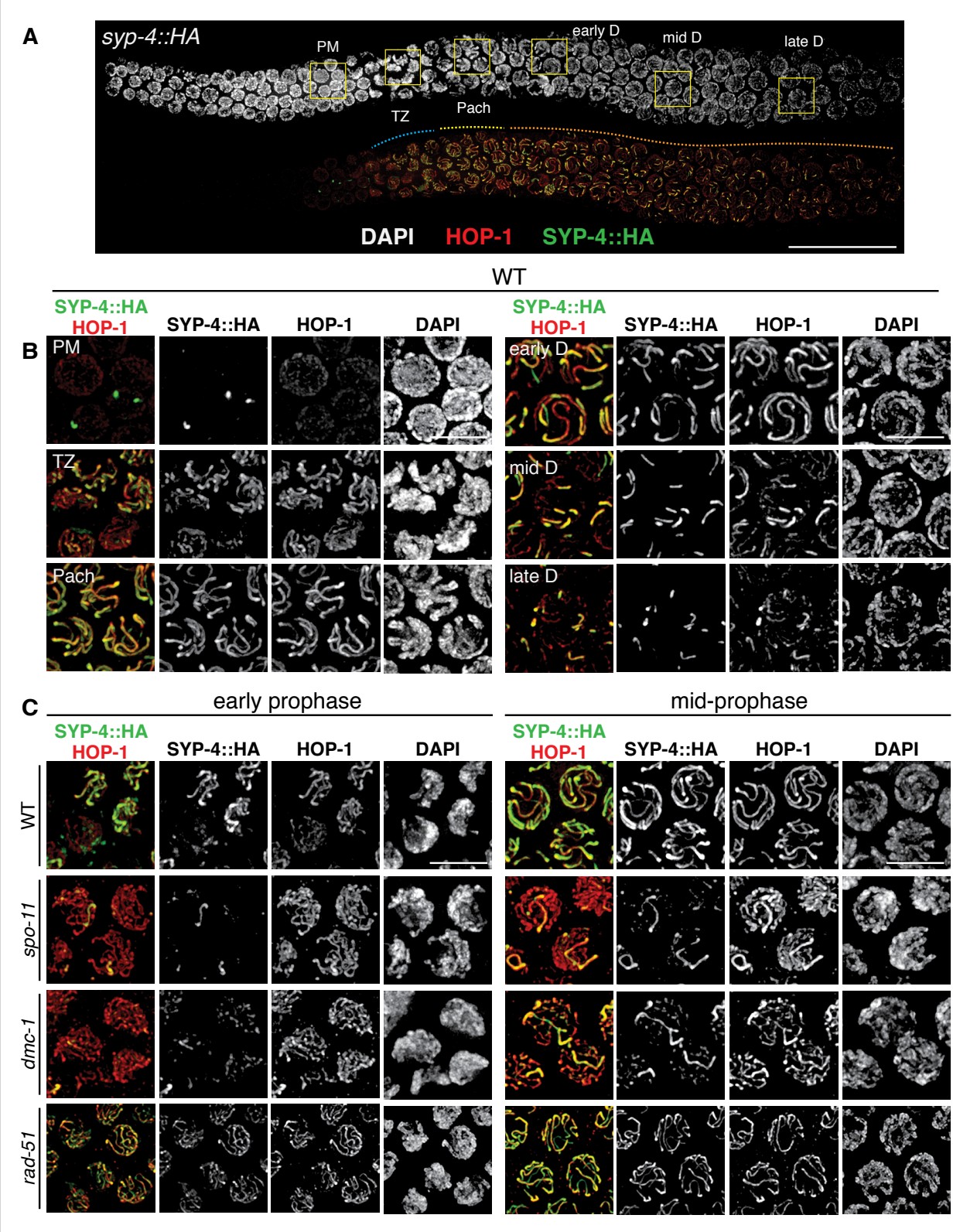

**Figure 3.** SPO-11 and DMC-1 are required for homologous synapsis in *P. pacificus*, while RAD-51 is dispensable. (**A**) Composite projection image of a wild-type strain expressing SYP-4::HA, stained with DAPI (gray), anti-HOP-1 (red), and anti-HA (green). Meiosis progresses from left to right. Scale bar, 30 μm. (**B**) Higher magnification images of wild-type nuclei in the premeiotic region (PM), transition zone (TZ), pachytene (Pach), and diplotene (**D**) stages. (**C**) Localization of SYP-4::HA and HOP-1 in WT, *spo-11*, *dmc-1*, and *rad-51* mutants during early and mid-prophase (roughly equivalent to the TZ

*Figure 3 continued on next page*

*Figure 3 continued*

and pachytene regions in wild-type germlines, respectively). Synapsis fails in the absence of *spo-11* and *dmc-1* function but occurs normally in *rad-51* mutants. Scale bar, 5 µm. See also *Figure 3—source data 1*.

The online version of this article includes the following source data for figure 3:

**Source data 1.** All epitope-tagged proteins described in this study support normal meiosis.

some late nuclei that retained clustered DAPI morphology, presumably either 'straggler' nuclei with delays in synapsis or CO designation, or apoptotic cells, both of which are typically observed in the germlines of wild-type *C. elegans* (*Figure 4D*). Differences between the localization of DMC-1 and RAD-51 were further validated by inserting a V5 epitope tag at the C-terminus of RAD-51. Staining of this strain with anti-V5 recapitulated the sparse, punctate localization seen with anti-RAD-51 antibodies, demonstrating that the broader distribution of DMC-1 is not an artifact of the V5 epitope tag or antibody (*Figure 4—figure supplement 1A*). RAD-51::V5 also supported normal meiosis, with a normal brood size, high embryonic viability, and low frequency of male self-progeny (*Figure 3—source data 1*). We also tagged DMC-1 at its C-terminus with an alternate epitope, 3xFLAG. While this tagged protein was not fully functional in homozygotes, we stained *dmc-1::3xflag/+* heterozygotes with anti-FLAG antibodies and observed a distribution indistinguishable from the DMC-1::V5 staining pattern (*Figure 4—figure supplement 2*).

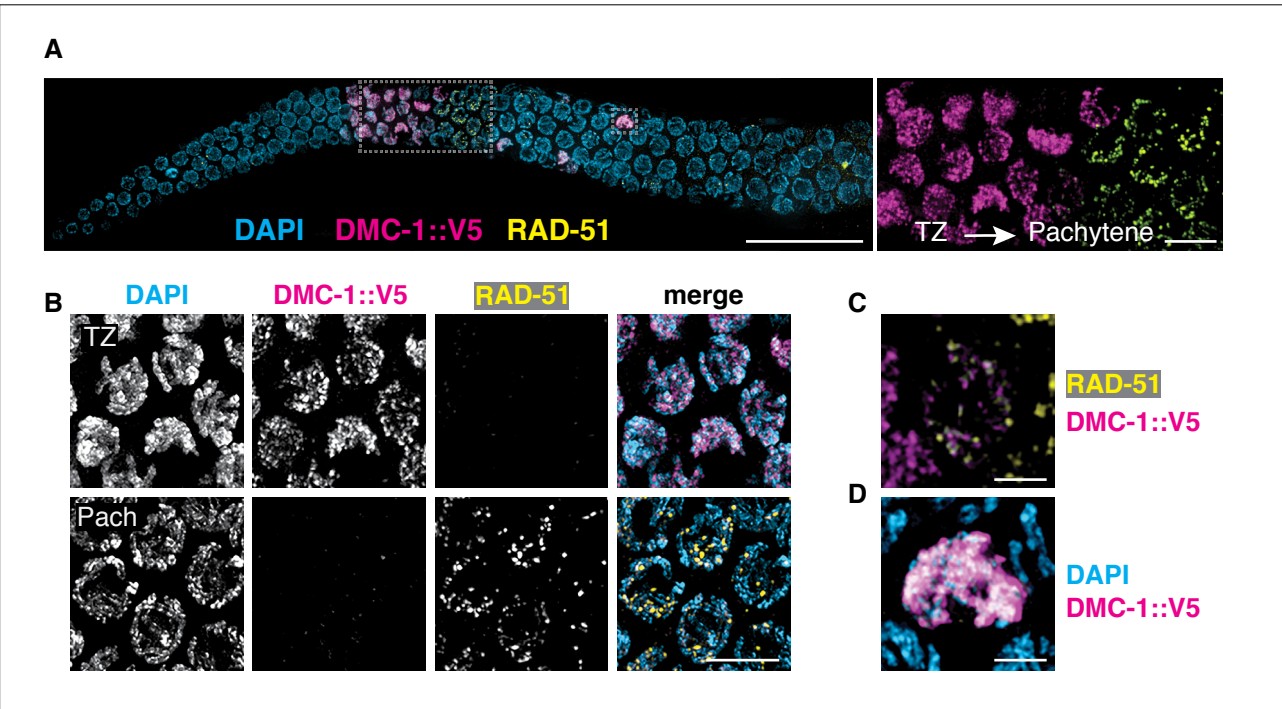

**Figure 4.** DMC-1 and RAD-51 localize sequentially to meiotic chromosomes. (**A**) Composite projection image of a wild-type gonad expressing DMC-1::V5, stained with DAPI (blue), anti-V5 (magenta), and anti-RAD-51 (yellow). Meiotic progression is from left to right. Scale bar, 30 µm. Inset shows the distinct localization of DMC-1 (magenta) and RAD-51 (yellow) in the transition zone and pachytene regions, respectively. Scale bar, 5 µm. (**B**) Higher magnification images of nuclei in the transition zone and pachytene region. DMC-1 is present along chromatin in the transition zone and disappears at pachytene. By contrast, RAD-51 localizes to discrete foci starting at pachytene. Scale bar, 5 µm. (**C**) Occasional nuclei at the transition from leptotene-zygotene to pachytene are positive for both DMC-1 and RAD-51. The signals do not completely overlap. Scale bar, 2 µm. (**D**) Example of a nucleus with polarized DAPI morphology and strong DMC-1 signal during later prophase. Such 'straggler' cells may be delayed in completing synapsis or undergoing apoptosis. Scale bar, 2 µm. See also *Figure 4—figure supplement 1*.

The online version of this article includes the following figure supplement(s) for figure 4:

**Figure supplement 1.** Localization of RAD-51:V5 using anti-V5 antibodies recapitulates the distribution of untagged RAD-51 detected with anti-RAD-51 polyclonal antibodies.

**Figure supplement 2.** Localization of DMC-1::3XFLAG in the germline of a hermaphrodite heterozygous for the epitope-tagged allele (*dmc-1::3xflag/+*).

We also tested the interdependence of DMC-1 and RAD-51 recombinases for their localization. In *S. cerevisiae* and *A. thaliana*, Dmc1 functions as an essential catalyst for interhomolog JM formation during meiotic DSB repair, while Rad51 acts as an accessory protein for Dmc1 nucleofilament formation (*Cloud et al., 2012*; *Da Ines et al., 2013*). We did not detect RAD-51 in transition zone nuclei, where DMC-1 was abundant on chromatin, and we found that DMC-1::V5 localization was normal in *rad-51* mutants, indicating that RAD-51 does not play an essential role in the recruitment of DMC-1 (*Figure 4—figure supplement 1B*). Conversely, in *dmc-1* mutants we detected RAD-51 foci only in late prophase nuclei, proximal to the very extended transition zone (*Figure 4—figure supplement 1C*). RAD-51 foci were more abundant and larger in *dmc-1* mutants than in wild-type pachytene nuclei, perhaps due to delays in repair due to the failure of homolog pairing and synapsis. Alternatively, the bright foci of RAD-51 observed in late prophase nuclei could reflect an apoptotic response to unrepaired breaks and/or extensive asynapsis.

The number of DMC-1 foci could not be quantified meaningfully due to their density and wide variation in intensity, but the broad distribution suggests that DMC-1 associates with intact dsDNA or chromatin-associated proteins, in addition to sites undergoing recombination. Nevertheless, in *spo-11* mutants DMC-1 was restricted to one or a few nuclear aggregates rather than broadly along chromosomes (*Figure 4—figure supplement 1D*). It was unclear whether these were associated with chromatin. This mislocalization may reflect either an absence of potential binding sites due to an absence of DSBs, or *spo-11*-dependent regulation of DMC-1 binding to chromatin, perhaps through activation of a DNA damage signaling pathway.

In contrast, RAD-51 foci are much sparser along chromosomes, suggesting that the protein localizes specifically to recombination intermediates. Thus, it was unsurprising to see that RAD-51 foci were absent from meiotic nuclei in *spo-11* mutants (*Figure 4—figure supplement 1F*). Some RAD-51 foci were observed in the mitotically proliferating region the germline in *spo-11* mutants, as in wildtype animals, providing a positive control for immunofluorescence (*Figure 4—figure supplement 1E*).

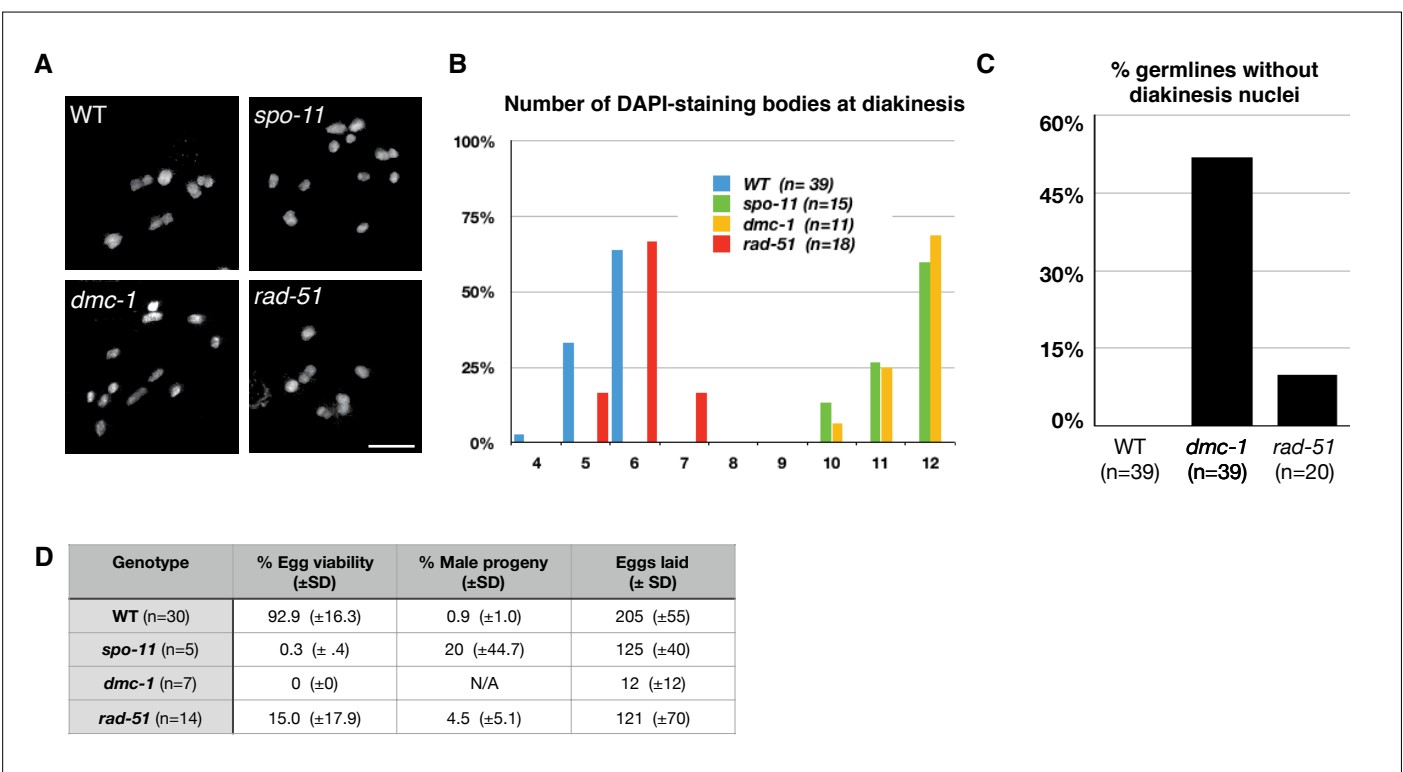

**Figure 5.** Crossover (CO) formation requires SPO-11 and DMC-1, but not RAD-51. (**A**) Representative images of DAPI-staining bodies at diakinesis for each indicated genotype. Scale bar, 5 µm. (**B**) Quantification of DAPI-staining bodies in the '–1' oocyte (immediately distal to the spermatheca) at diakinesis for each indicated genotype (n = represents number of nuclei scored). (**C**) Quantification of gonads that lacked nuclei with DAPI-staining bodies at diakinesis stage. *n* is the number of germlines scored for each genotype. (**D**) Frequencies of viable embryos and male progeny of whole broods from wild type, *spo-11*, *dmc-1*, and *rad-51* mutant hermaphrodites.

Together these observations indicate that DMC-1 and RAD-51 bind to chromatin at distinct stages of meiotic prophase and are not interdependent, although both require DSBs for their localization to chromosomes.

## RAD-51 is not required for CO formation or completion of meiosis

To assess the roles of DMC-1 and RAD-51 in CO formation, we quantified the number of DAPI-staining bodies at diakinesis in *dmc-1* and *rad-51* mutants. Wild-type oocytes at this stage usually have six bivalents that can be resolved as discrete DAPI-staining bodies (average = 5.6), while in *spo-11* mutants, ~12 DAPI-staining bodies were detected (average = 11.5), consistent with an absence of COs (*Figure 5A and B*). Interestingly, we frequently failed to detect oocytes at diakinesis in *dmc-1* mutant germlines, indicative of a defect in meiotic progression and the likely activation of a checkpoint in response to unrepaired DSBs. In cases when we did see nuclei at diakinesis, we observed an average of 11.6 DAPI-staining bodies, reflecting an absence of COs, as in *spo-11* mutants (*Figure 5A–C*).

Unexpectedly, disruption of *rad-51* resulted in homozygous mutant hermaphrodites that were viable and fertile, although animals produced smaller broods and their embryos showed greatly reduced viability, likely due to an inability to repair damage arising during DNA replication (*Figure 5D*). This was surprising because RAD-51 is essential for completion of meiosis in most organisms where it has been examined. Homozygous *rad-51* mutant gonads also displayed diakinesis nuclei more frequently than *dmc-1* mutants, although they were absent in 2 out of 20 gonads scored, indicating that loss of DMC-1 function impairs meiotic progression more severely than loss of RAD-51 (*Figure 5C*). Consistent with this observation, while self-fertilizing *rad-51* mutants had a lower average brood size than wild-type hermaphrodites, *dmc-1* mutants had even smaller broods, ranging from 0 to 35 embryos laid per mutant homozygote (*Figure 5D*). In striking contrast to *C. elegans rad-51* mutants, which display chromatin aggregates and fragments at diakinesis (*Martin et al., 2005*; *Rinaldo et al., 2002*), *Ppa-rad-51* mutants displayed an average of six DAPI-staining bodies, similar to wild-type (*Figure 5B*). Together with the relatively high viability of progeny of *rad-51* homozygous mutants, this indicates that RAD-51 does not play an essential role in CO formation in *P. pacificus*.

## COSA-1 marks designated CO sites during throughout early prophase

To further analyze CO formation in *P. pacificus*, we identified the gene encoding the metazoan meiotic cyclin-related protein COSA-1 (*C*rossover *S*ite *A*ssociated)/Cntd1 (Cyclin N-terminal Domain Containing 1; *Figure 1—figure supplement 8*) and inserted a 3xFLAG epitope tag at the C-terminus of the coding sequence. The strain expressing COSA-1::3xFLAG yielded progeny with high embryonic viability and few males, indicating that the tagged protein supports normal meiosis (*Figure 3—source data 1*). Immunostaining with anti-FLAG antibodies revealed discrete foci along the SC, beginning as early as zygotene, which decreased in number and became brighter within the short pachytene region (*Figure 6A and B*). Most pachytene nuclei displayed six COSA-1 foci, each of which was associated with an individual SC, indicating the presence of a single designated CO site between each pair of homologs (*Figure 6C* and *Figure 6—video 1*). We stained J4 larval-stage hermaphrodites whose germlines had not yet undergone the switch from spermatogenesis to oogenesis and found that pachytene spermatocytes also displayed approximately six COSA-1 foci (*Figure 6—figure supplement 1A*). This suggests that *P. pacificus*, like *C. elegans*, has robust chromosome-wide CO interference in both spermatogenesis and oogenesis.

Intriguingly, SC disassembly appeared to be regulated by the position of the designated CO site, as in *C. elegans*. By mid-prophase, six short stretches of SYP-4::HA were observed, each associated with a single COSA-1::3xFLAG focus near one end (*Figure 6B*). As meiosis progressed further, COSA-1::3xFLAG foci became undetectable, although short stretches of SYP-4::HA could still be observed. We also observed splaying of chromosome axes along the 'long arms' on one side of the COSA-1 focus upon disappearance of the SC from those regions (*Figure 6D*). HOP-1 was retained on both arms following SC disassembly, although the signal appeared fainter along the long arms, perhaps due to separation of the two axes. At this stage, short stretches of SYP-4::HA colocalize with corresponding bright stretches of HOP-1 (*Figures 3B and 6D*). Bivalents at diakinesis and meiotic metaphase I also displayed a cruciform structure similar to that seen in *C. elegans*, consistent with a single chiasma per chromosome pair (*Figure 1C*).

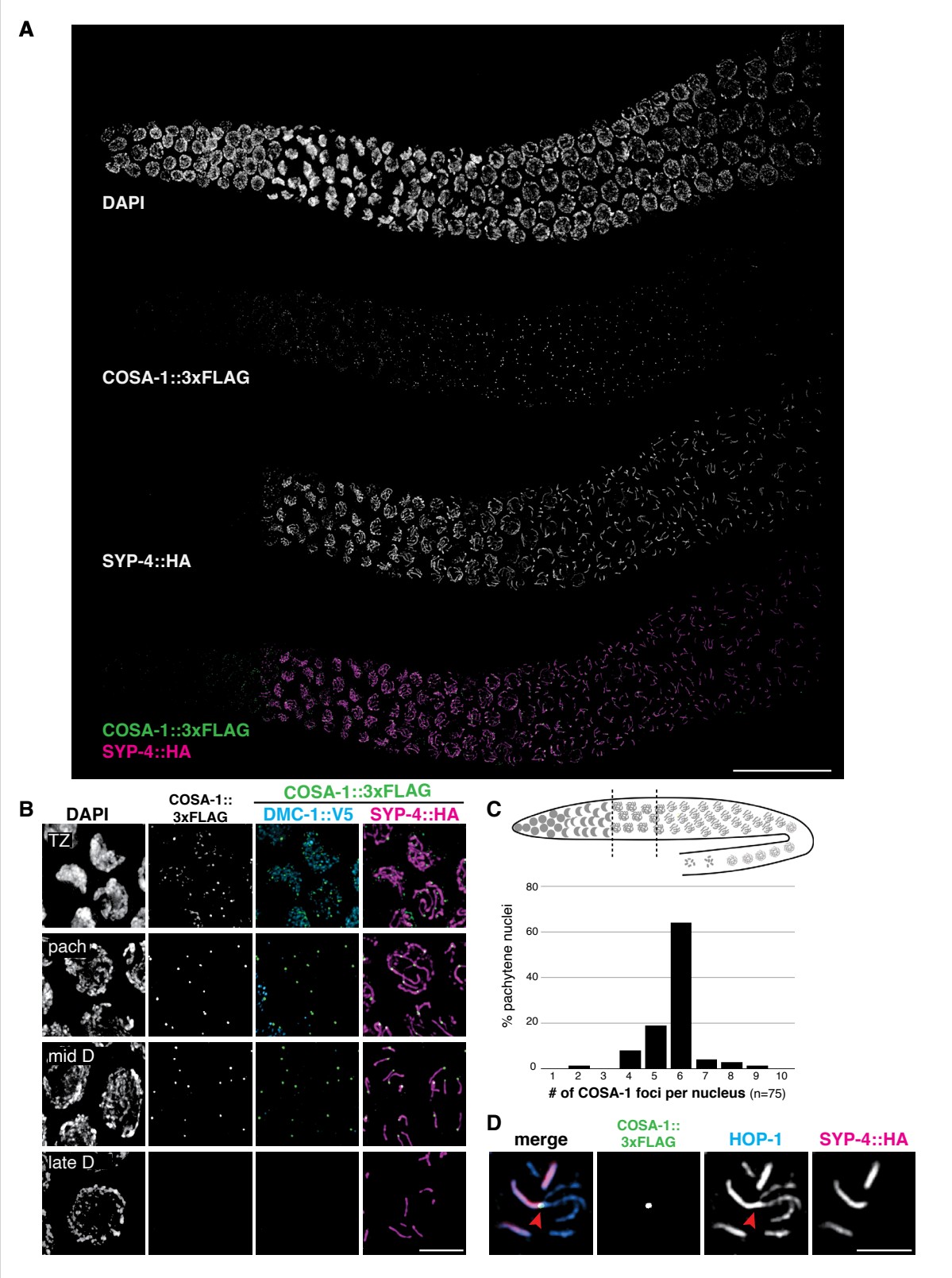

**Figure 6.** COSA-1 accumulates at a single focus per chromosome pair. (**A**) Composite projection image of a wild-type strain expressing three epitope-tagged proteins (COSA-1::3xFLAG, DMC-1::V5, and SYP-4::HA), stained with anti-FLAG, anti-V5, and anti-HA antibodies. Scale bar, 30 μm. (**B**) Higher magnification images of nuclei from the transition zone (TZ), pachytene (pach), mid- and late diplotene (**D**). COSA-1::3xFLAG (green) foci are detected in transition zone nuclei but do not colocalize with DMC-1::V5 (cyan). Foci peak in brightness in pachytene nuclei and gradually become dimmer until

*Figure 6 continued on next page*

*Figure 6 continued*

they are no longer detected during late diplotene. In early to mid-diplotene nuclei, six short stretches of SYP-4::HA (magenta) are observed per nucleus, each associated with a single COSA-1::3xFLAG focus. Scale bar, 5 µm. (**C**) Histogram showing the number of COSA-1::3xFLAG foci observed per nucleus in the pachytene region. Analysis was restricted to 15 nuclei per gonad immediately proximal to the transition zone and lacking DMC-1::V5 signal. Five individual gonads were analyzed, for a total of 75 nuclei scored. (**D**) Partial projection of a representative nucleus in mid to late diplotene, stained with anti-HOP-1 (blue), anti-HA (marking the synaptonemal complex [SC], magenta), and anti-FLAG (marking COSA-1, green). A single COSA-1::3xFLAG focus is observed at a junction (marked with a red arrowhead) between the 'short arm,' where SYP-4::HA is retained, and splayed 'long arms' lacking SC but positive for HOP-1. Scale bar, 2 µm. See also *Figure 6—figure supplement 1*, *Figure 6—video 1*, and *Figure 6—figure supplement 2*.

The online version of this article includes the following video and figure supplement(s) for figure 6:

**Figure supplement 1.** COSA-1 accumulates at a single site per chromosome pair during spermatogenesis.

**Figure supplement 2.** Asymmetric disassembly of the synaptonemal complex can occur along either arm of bivalent chromosomes.

**Figure 6—video 1.** COSA-1/Cntd1 accumulates at a single site per chromosome pair.

https://elifesciences.org/articles/70990/figures#fig6video1

In *C. elegans,* the orientation of chromosomes during meiotic segregation is stochastically determined by the position of the single CO between each pair: the 'long arm,' where SC first disassembles, is designated to retain cohesion and to lead towards the poles while the 'short arm' releases cohesion during MI, but then becomes the leading end for segregation of chromatids during meiosis II (*Albertson and Thomson, 1993*). Using FISH to mark specific chromosome regions together with immunostaining of SYP-4::HA, we found that the same was true for *P. pacificus*: an asymmetrically localized probe localized to either the long (desynapsed) arm or the short arm in different nuclei (*Figure 6—figure supplement 2*).

We examined the localization of COSA-1::3xFLAG in various mutant backgrounds. As expected, *dmc-1* mutants showed a complete absence of COSA-1 foci throughout prophase, while six foci were observed in pachytene nuclei in *rad-51* mutants (*Figure 7A*), consistent with the number of DAPI-staining bodies observed at diakinesis in these mutants (*Figure 5B*). A few bright COSA-1::3xFLAG foci were present throughout prophase in *spo-11* mutants (*Figure 7A*). However, since ~12 DAPI-staining bodies were observed during diakinesis, we conclude that these COSA-1 foci do not mark designated COs. A similar phenomenon has been reported in *C. elegans spo-11* mutants (*Nadarajan et al., 2017*; *Pattabiraman et al., 2017*), suggesting that COSA-1 and other CO factors can coalesce at sites lacking bona fide recombination intermediates when such structures are absent.

## A genetic map for *P. pacificus* reveals conservation of the CO landscape

Prior work has led to divergent estimates of the meiotic recombination frequency in *P. pacificus*. By summing over measured genetic intervals on the same linkage group, the Sommer lab initially concluded that the map length of each chromosome exceeded 100 cM, with a maximum length of 215 cM for chromosome I, which is also physically the longest (*Hong and Sommer, 2006*; *Srinivasan et al., 2003*; *Srinivasan et al., 2002*). However, a map derived by the same group based on RNA sequencing of~ F10 recombinant inbred lines (RILs) yielded markedly shorter genetic lengths, below 100 cM per chromosome (*Rödelsperger et al., 2017*).

Since an understanding of the CO landscape is an important reference for analysis of meiosis, we addressed this ambiguity by constructing a new genetic map for *P. pacificus*. We used three divergent parental strains from different geographic regions: PS312 from California (CA), PS1843 from Washington State (WA), and RSB001 from La Réunion island (LR), and mated them using a double-cross hybrid strategy. A total of 93 progeny from a cross between CA/WA hybrid males and CA/LR hybrid hermaphrodites were sequenced (*Figure 8A*). This strategy allowed us to simultaneously map COs that occur during hermaphrodite oogenesis and male spermatogenesis from the same progeny (*Figure 8B*).

We draw several conclusions from the resulting CO maps (*Figure 8C*). First, the map lengths for each chromosome are close to 50 cM for both oocyte and spermatocyte meiosis, very similar to genetic map lengths in *C. elegans*. This indicates that a single CO usually occurs between each pair of chromosomes per meiosis, consistent with our cytological observations. Close to half of all chromatids inherited from each parent were nonrecombinant as expected if a single CO occurs between two of the four possible chromatids. We did observe a few examples of double COs in our data, but such events were rare (6/1012 chromatids), and were most prevalent on the longest chromosome (chromosome

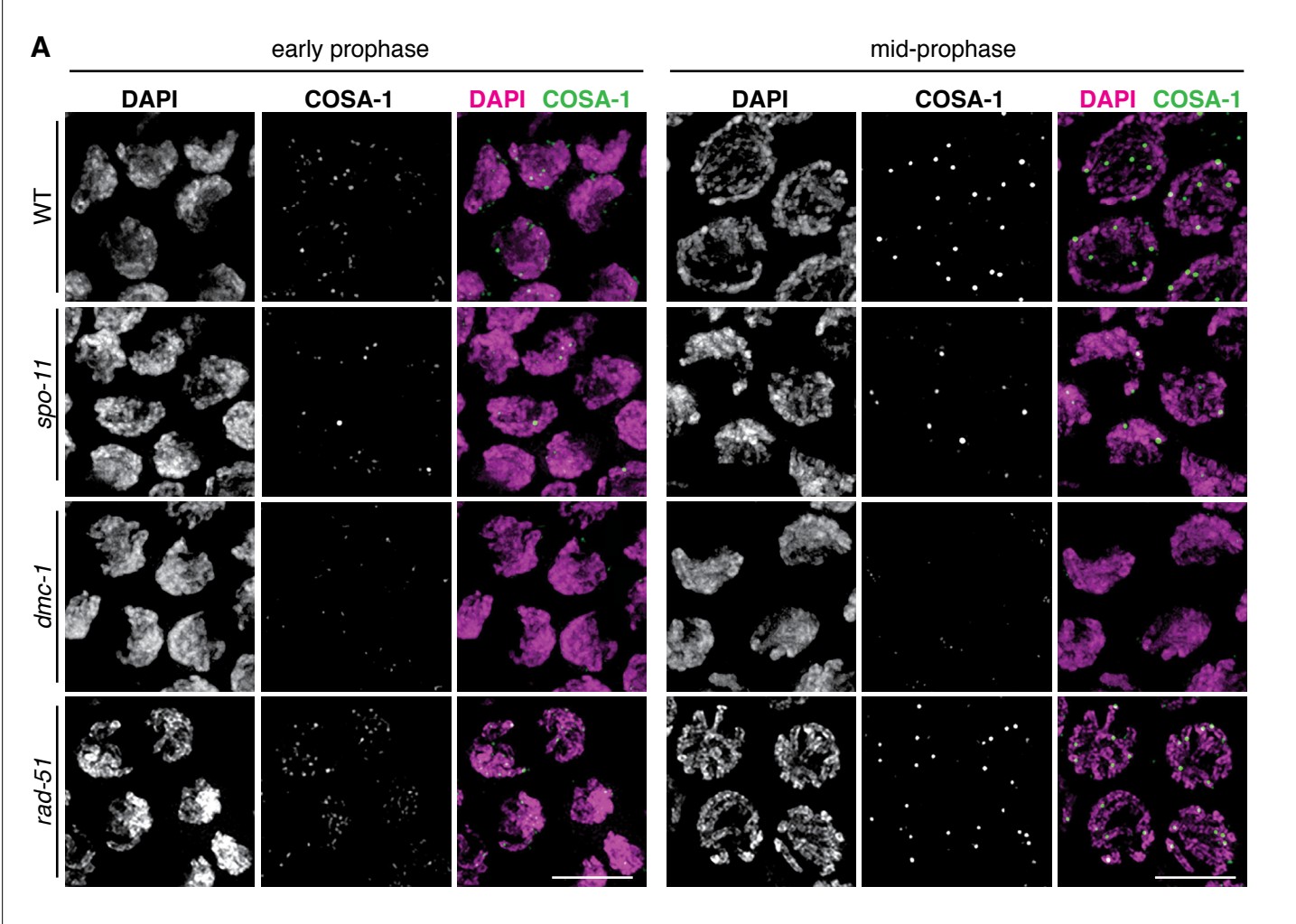

**Figure 7.** COSA-1::3xFLAG accumulates at sites of presumptive crossovers (COs). (**A**) Nuclei from hermaphrodites of the indicated genotype displaying COSA-1::3xFLAG (green) in early and mid-prophase (roughly equivalent to the transition zone and pachytene regions in wild-type germlines, respectively). COSA-1 foci are absent in *dmc-1* mutants, but six foci per nucleus are detected in wild-type and *rad-51* mutants. Occasional foci are detected in s*po-11* mutants. Scale bar, 5 μm.

I; *Supplementary file 5*). Second, the distal regions ('arms') show higher recombination rates than the central regions of each chromosome, as in *Caenorhabditis* (*Barnes et al., 1995*; *Rockman and Kruglyak, 2009*). This bias was also evident in a map constructed using RNA sequencing data (*Rödelsperger et al., 2017*) and seems to be a widely conserved feature among Rhabditids and perhaps other nematode clades (*Doyle et al., 2018*; *Gonzalez de la Rosa et al., 2021*).

Third, the male spermatocyte map showed a pronounced asymmetry for chromosomes III and IV, with COs occurring predominantly on one arm, whereas the corresponding oocyte maps show COs on both arms. Conversely, the oocyte-specific map for the X chromosome shows an absence of COs on the right arm (the single X chromosome does not undergo COs during male spermatogenesis; *Figure 8C*). These findings suggest that COs were suppressed on one side of these chromosomes in the hybrid parents. This suppression may reflect structural rearrangements between the parental strains. Specifically, we predict that there are structural differences between the Washington and California versions of chromosomes III and IV, and between the La Réunion and California versions of the X chromosome. A lack of COs on the right arm of the X chromosome was also evident in data from RILs from a cross between the same strains from La Réunion and California (*Rödelsperger et al., 2017*). The CO suppression on chromosomes III and IV in WA/CA hybrid spermatocytes likely reflects

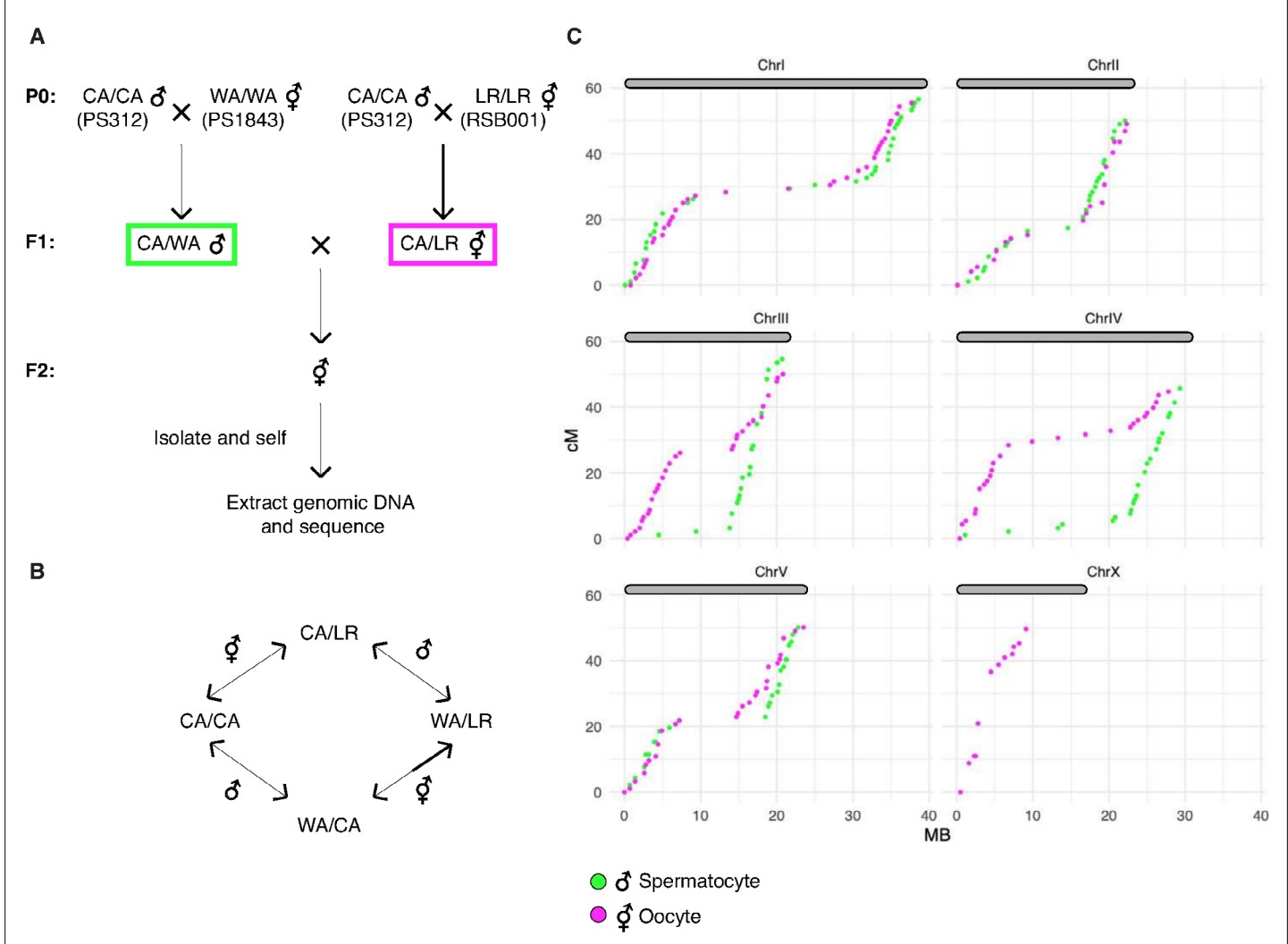

**Figure 8.** A genetic map for *Pristionchus pacificus* based on recombination in inter-strain hybrids. (**A**) Crossing scheme to generate a recombination map using three parental strains. California PS312 (CA), Washington PS1843 (WA), and La Réunion Island RSB001 (LR) strains were crossed to obtain F1 hybrids, which were then crossed to each other. Whole-genome sequencing of progeny from crosses between hybrid F1s enabled the analysis of meiotic recombination events in each F1 parent. (**B**) Genotype transitions along a chromosome in F2 correspond to recombination in the male or hermaphrodite F1 parent. (**C**) Marey plots show genetic map position in centimorgans vs. the physical position in megabases for male (green) and hermaphrodite (magenta) meiosis. Each bin was treated as a single locus and dots were plotted at the center of each marker bin. Map positions were computed with OneMap as described. The observed map length of ~50 cM indicates that chromosomes undergo an average of one crossover (CO) per meiosis. The X chromosome lacks a homolog in males, so there is no male-specific map for the X in our data. See also *Figure 8—figure supplement 1* and *Figure 8—source data 1*.

The online version of this article includes the following source data and figure supplement(s) for figure 8:

**Source data 1.** Genotype calls and maps distances plotted in *Figure 8* and *Figure 8—figure supplement 1*.

**Figure supplement 1.** Identification of informative markers and genotyping of progeny from hybrid animals.

intra- rather than inter-chromosomal rearrangements since the data do not show pseudolinkage of loci on these chromosomes, as would be expected if a III; IV translocation were present.

Notably, the asymmetry in CO suppression also suggests that one arm of each chromosome may contain a region that plays a dominant role in CO formation during meiosis, perhaps related to the function of PCs in *C. elegans* (see Discussion for more details). This was unexpected given the absence of obvious homologs of the zinc finger proteins required for PC function in *C. elegans*, and the more 'conventional' dependence of pairing and synapsis on recombination that we have documented for *P. pacificus* (see Discussion).

Finally, we detected pronounced segregation distortion for chromosomes I and II in the oocyte map, with inheritance of the La Réunion haplotype strongly favored (*Figure 8—figure supplement 1B*). In each case nonrecombinant chromosomes of the La Réunion haplotype were inherited about eightfold more frequently than the nonrecombinant California chromatids. These observations are consistent with either a bias in meiosis segregation – that is, meiotic drive – or the presence of loci on these chromosomes that cause genetic incompatibilities (i.e., inviability of some F2 progeny). Analysis of recombinant chromatids suggests that the latter scenario is more likely since a gradient in the severity of the distortion can be detected along the physical length of the chromosomes, consistent with the idea that specific loci on the affected autosomes may be toxic to hybrid progeny; it is more difficult to imagine how specific loci could lead to meiotic drive in a holocentric species.

## Discussion

### Distinct roles for DMC-1 and RAD-51

Comparison of the activities of Rad51 and Dmc1 in vitro has revealed similar profiles: both RecA homologs bind preferentially to single-stranded DNA and can mediate strand exchange reactions. However, Dmc1 is uniquely required during meiosis. The distinct requirements for Rad51 and Dmc1 are thought to be due in part to the activity of Dmc1-specific cofactors Mnd1 and Hop2, which confer different activities, and/or a higher tolerance for mismatches by Dmc1, which may enable it to promote recombination between nonidentical homologs (*Steinfeld et al., 2019*).

Our analysis of RAD-51 and DMC-1 in *P. pacificus* reveals their distinct contributions during meiosis. The binding of these proteins to chromatin appears very different; the abundance of DMC-1 foci suggests that it binds to sites other than repair intermediates. In contrast, RAD-51 displays a more restricted punctate localization, and only after DMC-1 is largely removed from chromosomes upon completion of synapsis. Intriguingly, both DMC-1 and RAD-51 depend on SPO-11 for their association with meiotic chromosomes. We interpret these observations to indicate that DMC-1 is positively regulated by a mechanism that responds to SPO-11 activity, likely through activation of a DNA damage signaling pathway. In contrast, the association of RAD-51 with recombination intermediates may normally be inhibited until after the establishment of CO intermediates between each pair of chromosomes.

The sequential localization of DMC-1 and RAD-51 first suggested that they function independently, and this is further supported by our analysis of loss-of-function mutations. In contrast to budding yeast and *A. thaliana*, we have found that RAD-51 in *P. pacificus* is dispensable for the activity of DMC-1 in pairing, synapsis, and CO formation. Instead, RAD-51 appears to play a supporting role in DSB repair during pachytene, promoting repair of DSBs that remain after CO designation has occurred. In *C. elegans*, which expresses only RAD-51, a similar switch between two modes of DSB repair is nevertheless observed during meiotic prophase: association of RAD-51 with repair intermediates is differentially regulated from the onset of meiosis until a mid-pachytene transition that coincides with CO designation; at this time, competence to convert DSBs to interhomolog COs is also lost (*Hayashi et al., 2007*). An analogous switch from a 'meiotic' repair to a 'somatic'-like repair pathway in mid-pachytene has also been described in mouse spermatocytes (*Enguita-Marruedo et al., 2019*).

Taken together, it appears that DMC-1 and RAD-51 have specialized functions in *P. pacificus*: formation of interhomolog CO intermediates by DMC-1 prior to completion of synapsis, followed by a more generic mode of DSB repair mediated by RAD-51. Our observations that nuclei in *rad-51* mutants display cruciform bivalents and lack fragmented chromatin at diakinesis suggest that excess DSBs can be repaired through an alternate pathway that does not depend on RAD-51 activity, such as non-homologous end joining, or that DMC-1 can compensate for loss of RAD-51 but not vice versa. Future studies may reveal how the activities of RAD-51 and DMC-1 are regulated to accomplish an orderly hand-off during meiotic prophase.

### Comparative analysis of meiosis reveals variations and similarities within the nematode lineage

In addition to establishing key aspects of meiosis in *P. pacificus*, this work also illuminates the evolutionary history of meiosis in *C. elegans*. A body of prior work has revealed that recombination-independent homologous synapsis in *C. elegans* relies on PCs, specialized chromosome regions that

interact with nuclear envelope and drive chromosome movement during early prophase. Similar roles in chromosome movement and pairing during meiosis are typically mediated by telomeres, but have shifted to a unique region on each chromosome in *C. elegans*. PCs in *C. elegans* mediate synapsis-independent pairing and also act as the sites of synapsis initiation (*MacQueen et al., 2005*; *Rog and Dernburg, 2013*). In most other organisms, telomere-led chromosome movement is thought to promote homologous interactions, but stabilization of pairing and initiation of synapsis require and often occur at early recombination intermediates, which depend on Spo11 and Dmc1.

PC activity in *C. elegans* depends on and is largely defined by the recruitment of a family of zinc finger proteins that bind to DNA sequence motifs in these regions (*Phillips et al., 2009b*). These proteins, known as ZIM-1, ZIM-2, ZIM-3, and HIM-8 in *C. elegans*, act as scaffolds to recruit a cascade of kinase activities required for pairing and synapsis (*Harper et al., 2011*; *Kim et al., 2015*; *Labella et al., 2011*). PCs have been regarded as an evolutionary alternative to 'canonical' Dmc1-mediated pairing mechanisms, so we found it intriguing that many of the sequenced genomes of nematodes in Clade V include homologs of the HIM-8/ZIM family as well as Dmc1, Hop2, and Mnd1 (*Figure 2—figure supplement 1*). The *Pristionchus* genus is unusual within this Clade in that it lacks apparent homologs of the PC proteins, while *Caenorhabditis* is among the few genera that lack Dmc1/Mnd1/Hop2, although these genes were independently lost along the branch leading to *Oscheius tipulae*. Because these homologs are present in species that retain intact DMC-1 genes, it is unlikely that PC proteins arose as an alternative to DMC-1 function in meiosis, although in *Caenorhabditis* they may have acquired a synapsis initiation activity that eventually allowed degeneration and loss of DMC-1. Our phylogenetic analysis suggests that the ability of PCs to promote homologous synapsis in the absence of DSBs and strand invasion may be recently derived, and perhaps restricted to *Caenorhabditis*.

The existence of PCs in *C. elegans* was initially inferred because chromosome rearrangements often result in asymmetric CO suppression: regions distal to translocation breakpoints are inhibited from crossing over, while the chromosome region containing the PC undergoes elevated recombination due to a feedback mechanism that helps to ensure an 'obligate crossover' (*Rose and McKIM, 1992*; *Yu et al., 2016*). The asymmetric CO suppression we detected in our genetic map data suggests that a directional CO-promoting activity may also be located near one end of each chromosome in *P. pacificus*. This was unexpected given the absence of obvious homologs of the zinc finger proteins required for PC function in *C. elegans*, and the more 'conventional' dependence of pairing and synapsis on recombination that we have documented for *P. pacificus*. Alternatively, the CO suppression we observed may result from large inversions that suppress homologous pairing or synapsis along entire arms of several chromosomes. However, such large-scale structural heterozygosity has not been detected within other nematode species and would likely result in complete reproductive isolation, so we favor the idea that local interruptions in sequence contiguity may lead to CO suppression due to a CO-promoting activity that acts directionally from one end of each chromosome. Complete genome assemblies of different *P. pacificus* strains will help to resolve this issue.

If PCs are indeed conserved in *P. pacificus*, they clearly do not mediate recombination-independent synapsis, as in *C. elegans*. Based on our cytological observations, it is more likely that synapsis initiates at recombination intermediates in *P. pacificus*, although this has not yet been tested rigorously. We speculate that PCs may act as a source of a protein factor required for CO designation, and that this factor must move from the source to a recombination intermediate along the SC. For example, in mammals Cdk2 is initially found at telomeres and later at CO intermediates; perhaps it moves along the SC, thereby ensuring that only recombination intermediates linked in cis to homologous telomeres can eventually become COs. For this hypothetical movement of molecules to enable CO designation, the SC must span the distance between the 'source' and a recombination intermediate, but assembly of the SC could initiate at either site.

We have previously speculated that unique PC regions may have arisen in holocentric nematodes as part of a meiotic system that promotes karyotype stability (*MacQueen et al., 2005*; *Rog and Dernburg, 2013*), an idea that seems more viable in light of our indirect evidence for their conservation in *Pristionchus*. The existence of such a mechanism could potentially account for the preservation of genetic linkage units known as 'Nigon elements' over long evolutionary periods, which has been documented through genome sequencing of diverse nematodes (*Foster et al., 2020*; *Gonzalez de la Rosa et al., 2021*; *Tandonnet et al., 2019*). When inversions arise that suppress local recombination,

they can lead to intraspecies reproductive barriers such as toxin/antitoxin systems, and eventually to full speciation.

*P. pacificus* shares many experimental advantages with *C. elegans* for molecular studies of meiosis. Naturally, it offers a more limited suite of experimental tools since it was more recently developed and is less widely studied. Genome engineering using CRISPR/Cas9 is less efficient than in *C. elegans*, and has so far been refractory to large insertions, such as transgenes encoding fluorescent proteins. However, this barrier will likely be overcome, just as technical tweaks have greatly improved editing efficiency in *C. elegans* in less than a decade. Another obstacle has been the absence of balancer chromosomes, which are invaluable for maintaining mutations in essential genes in *C. elegans*. Notably, our genetic map data indicating the likely presence of CO-suppressing rearrangements between divergent strains of *P. pacificus* suggests that it may be possible to propagate balancer chromosomes in this species, which would be welcome additions to the experimental toolbox.

Our work has now established that the recombination landscape is remarkably well conserved between *P. pacificus* and *C. elegans*. This was initially surprising in light of the major differences in their regulation of synapsis and recombination. Indeed, comparison of *P. pacificus* and *C. elegans* reveals that enrichment of COs on the chromosome arms and robust chromosome-wide CO interference can be implemented through divergent meiotic programs, for example, with synapsis either dependent on or independent of CO precursors, and with CO sites designated either before or after completion of synapsis. Interestingly, other features of chromosome organization, including enrichment for novel and developmentally regulated genes on the arms, with more ancestral and broadly expressed genes concentrated in the centers, as well as chromosome-wide chromatin modification profiles, are also conserved across the Rhabditina (*Werner et al., 2018*). How the recombination landscape is shaped by and/or contributes to these genomic features is not well understood, but the conservation of these features over tens of millions of years of evolution suggests that they are important for genome transmission and/or function.

Nematodes are an ancient and widespread phylum in which most and perhaps all species have holocentric chromosomes. The presence of spindle-attachment sites along the length of chromosomes poses unique challenges for both mitosis and meiosis (*Dernburg, 2001*; *Melters et al., 2012*). In monocentric organisms, the centromeres and/or pericentromeric regions typically recruit factors such as Shugoshin to maintain cohesion between sister chromatids during the first, reductional meiotic division, while release of cohesion along chromosome arms allows segregation of recombinant homologs (*Watanabe and Kitajima, 2005*). Holocentric chromosomes must use alternate mechanisms to implement the stepwise loss of cohesion needed for two orderly meiotic divisions. The asymmetric SC disassembly we observe in *P. pacificus*, together with our evidence that the orientation of chromosomes during meiotic divisions is stochastically determined by CO position (*Figure 6—figure supplement 2*) likely reflects conservation of a mechanism discovered in *C. elegans* in which CO designation leads to asymmetric SC disassembly, retention of cohesion on one side of the CO site during the first division, and release on the other side (*Ferrandiz et al., 2018*; *Martinez-Perez et al., 2008*; *Severson and Meyer, 2014*; *Tzur et al., 2012*). Thus, this particular solution to the problem of segregating holocentric chromosomes in meiosis appears to predate the Rhabditine radiation. This pathway relies on the formation of two bivalent 'arms' with reciprocal functions, which in turn depends on strict CO interference; multiple COs 'confuse' the system and lead to frequent missegregation (*Hollis et al., 2020*). We expect that these two features of meiosis are widely shared among and perhaps beyond Clade V nematodes. However, they may not be universal among nematodes; for example, studies in Ascarids have shown that heterochromatic regions near one end of a chromosome act as meiotic centromeres and are eliminated in somatic cells (*Goday and Pimpinelli, 1989*; *Melters et al., 2012*). This mechanism may allow for more relaxed CO control and/or a different pattern of COs between homologous chromosomes.

Inexpensive genome sequencing and the adaptability of CRISPR/Cas-based genome editing has enabled rapid progress in developing new experimental models such as *P. pacificus* for molecular studies. Future studies in *P. pacificus* and other nematodes will likely expand our understanding of core mechanisms and plasticity of sexual reproduction.

## Materials and methods

**Key resources table**

| Reagent type (species) or resource | Designation | Source or reference | Identifiers | Additional information |
|---|---|---|---|---|
| Gene (*Pristionchus pacificus*) | *cenp-c* | El Paco genome reference, v1, 2017/Wormbase/this paper | El Paco genome reference, v1, 2017 ID:UMMS71-6.7mRNA-1; Wormbase ID:PPA37734 | See *Supplementary file 1* |
| Gene (*Pristionchus pacificus*) | *cosa-1* | El Paco genome reference, v1, 2017/Wormbase/this paper | El Paco genome reference, v1, 2017 ID:UMMS57-3.22mRNA-1; Wormbase ID:PPA23791 | See *Supplementary file 1* |
| Gene (*Pristionchus pacificus*) | *dmc-1* | El Paco genome reference, v1, 2017/this paper | El Paco genome reference, v1, 2017 ID:UMMS442-1.74mRNA-1 | See *Supplementary file 1* |
| Gene (*Pristionchus pacificus*) | *hop-1* | El Paco genome reference, v1, 2017/Wormbase/this paper | El Paco genome reference, v1, 2017 ID:UMMS341-6.31mRNA-1; Wormbase ID:PPA10281 | See *Supplementary file 1* |
| Gene (*Pristionchus pacificus*) | *rad-51* | El Paco genome reference, v1, 2017/Wormbase/this paper | El Paco genome reference, v1, 2017 ID:UMMS442-1.74mRNA-1; Wormbase ID:PPA42255 | See *Supplementary file 1* |
| Gene (*Pristionchus pacificus*) | *spo-11* | El Paco genome reference, v1, 2017/Wormbase/this paper | El Paco genome reference, v1, 2017 ID: UMMS230-10.9mRNA-1; Wormbase ID:PPA33054 | See *Supplementary file 1* |
| Gene (*Pristionchus pacificus*) | *syp-4* | El Paco genome reference, v1, 2017/this paper | El Paco genome reference, v1, 2017 ID: UMMS245-8.16mRNA-1 | See *Supplementary file 1* |
| Strain, strain background (*Pristionchus pacificus*) | PS312, isolate 97 | Sommer Lab, MPI | | |
| Strain, strain background (*Pristionchus pacificus*) | PS1843 | Sommer Lab, MPI | | |
| Strain, strain background (*Pristionchus pacificus*) | RSB001 | Sommer Lab, MPI | | |
| Strain, strain background (*Pristionchus pacificus*) | *P. pacificus* allele and strain information | This paper | N/A | See *Supplementary file 1* |
| Antibody | (Mouse polyclonal) anti-Ppa-RAD-51 | Pocono Rabbit Farm and Laboratory, Canadensis, PA | | (1:300) |
| Antibody | (Rabbit polyclonal) anti-Ppa-HOP-1 | SDIX, Newark, DE | | (1:300) |
| Antibody | (Mouse monoclonal) anti-FLAG M2 | Millipore Sigma | Cat#F1804; RRID:AB_262044 | (1:500) |
| Antibody | (Mouse monoclonal) anti-V5 | Thermo Fisher Scientific | Cat#R960-25; RRID:AB_2556564 | (1:500) |
| Antibody | (Rabbit polyclonal) anti-V5 | Millipore Sigma | Cat#V8137; RRID:AB_261889 | (1:250) |
| Antibody | (Goat polyclonal) anti-HA | Novus Biologicals | Cat##NB600-362; RRID:AB_10124937 | (1:500) |
| Antibody | (Mouse monoclonal) anti-a-tubulin, clone DM1A | Millipore Sigma | Cat#05-829; RRID:AB_310035 | (1:400) |
| Sequence-based reagent | FISH probe to the center of Chromosome IV | This paper | N/A | See Materials and methods, FISH probes |
| Sequence-based reagent | FISH probe to the left end of chromosone X | This paper | N/A | See Materials and methods, FISH probes |
| Sequence-based reagent | Alt-R CRISPR tracrRNA | Integrated DNA Technologies, Coralville, IA | Cat#1072534 | |

| Reagent type (species) or resource | Designation | Source or reference | Identifiers | Additional information |
|---|---|---|---|---|
| Sequence-based reagent | Guide RNAs, DNA repair templates and genotyping primers | This paper | N/A | See *Supplementary file 1* |
| Peptide, recombinant protein | *Streptococcus pyogenes* Cas9-NLS purified protein | QB3 MacroLab at UC Berkeley | N/A | |
| Commercial assay or kit | DNeasy Blood & Tissue Kit | Qiagen | Cat#69504 | |
| Commercial assay or kit | plexWell LP384 Library Preparation Kit | seqWell | LP384 | |
| Software, algorithm | For a list of used software and relevant parameters, see *Supplementary file 4* | | | |

## *P. pacificus* strains and maintenance

Animals were cultured on NGM media with *Escherichia coli* OP50 at 20 °C under the standard conditions developed for *C. elegans* (*Brenner, 1974*). The wild-type strain in which most mutations were generated was a derivative of PS312 designated as '97.' The Sommer lab found that this isolate was more amenable to genome editing by CRISPR-Cas9 than the parental strain. Mutant alleles were maintained in unbalanced heterozygotes. Every few generations and immediately prior to any analysis, single adult hermaphrodites were picked to new plates and allowed to lay embryos for 2 days, after which the genotype of the parent was determined by PCR genotyping. Progeny from heterozygous mothers, 25% of which are homozygous for the meiotic mutation, were analyzed using the assays described here; heterozygous and wild-type siblings were frequently used as controls, in addition to unedited wild-type animals.

## CRISPR/Cas9-mediated genome editing

To modify the *P. pacificus* genome, we adapted our preferred CRISPR/Cas9 protocol from *C. elegans* to *P. pacificus*. Equimolar quantities of Alt-RCRISPR-Cas9 crRNA and tracrRNA molecules (Integrated DNA Technologies, Coralville, IA) were hybridized using a thermocycler (95 °C for 5 min, then 25 °C for 5 min). 4 µl of 100 µM hybridized tracrRNA/crRNA was combined with 4 µl of 40 µM *Streptococcus pyogenes* Cas9-NLS purified protein (QB3 MacroLab, UC Berkeley, Berkeley, CA) and incubated at room temperature for 5 min. 2 µl of 100 µM stock of an Ultramer DNA oligo (IDT) repair template containing 50–60 bp homology arms and the desired epitope or mutation sequence was added to the mixture, for a total volume of 10 µl, and injected into the gonads of young adult hermaphrodites aged 24 hr from the final larval (J4) stage. Following a 2–4 hr recovery period, injected animals were allowed to lay embryos at 20 °C for 16–20 hr. 4 days later, a fraction of the F1 population (typically 150–200 progeny from 6 to 8 injected $P_0$s) was screened for the presence of the desired mutation or epitope tag sequence by PCR, and candidate alleles were verified by Sanger sequencing. A complete list of crRNA, repair template, and genotyping primer sequences used to generate alleles in this study is provided as *Supplementary file 1*.

TALEN constructs used in our initial analysis of *dmc-1* and *spo-11* were generated using a protocol adapted from *Zhang et al., 2011* and designed using the TAL Effector Nucleotide Targeter 2.0 website (https://tale-nt.cac.cornell.edu/).

## Brood analysis

To quantify embryonic viability, brood size, and male progeny of wild-type and *rad-51* mutants, J4 (virgin) hermaphrodites were picked onto individual plates and transferred every 24 hr over 72 hr total. Embryos were counted each day, after transferring the adult hermaphrodite to a new plate, and kept at 20 °C. 3–4 days later, adults were counted on each plate. To analyze *spo-11* and *dmc-1* mutants, 24 individual J4 hermaphrodites were picked from progeny of a verified heterozygous mutant hermaphrodite. Quantification was performed as in wild type, but after 72 hr, the adult hermaphrodite was lysed and genotyped for the presence of the mutation. Thus, although 24 animals total were quantified from a mixed population of *spo-11/+* or *dmc-1/+* animals, data from five homozygous *spo-11* and 7 homozygous *dmc-1* mutant animals are reported in the data table (*Figure 5D*).

## FISH probes

Probes targeting a central locus on chromosome IV and the left end of chromosome X were designed based on two short tandem repeat motifs. Tandem Repeat Finder v4.09 (*Benson, 1999*) was used to identify tandem repeats in *P. pacificus* 'El Paco' genome assembly (*Rödelsperger et al., 2017*) using default parameters and a maximum periodicity of 200 bp. The output was then filtered to identify repeats that spanned more than 8 kb. These were compared to the genome sequence using BLAST to identify the subset of sequences restricted to a single major locus per genome. A subset of these repeats was then tested for specific and robust hybridization with oligonucleotide probes. The chromosome IV probe (TCATTGAAATGATCACAATCATTGA) targets a 30-base repeated motif ATGATCAT TGAAATGATCACAATCATTGAG, which spans 40.1 kb at a position 11.3 Mb from the left end of chromosome IV. The chromosome X probe (GGTGGTCGACGGCTGCGTCG) targets the 30-base repeat motif GGTGGTCGACGGCTGCGTCGACTGAAGAGT that spans two very close regions of 29.3 kb and 11.1 kb on the left end of the X chromosome. Single-stranded oligonucleotides labeled at the 3' end with 6-FAM or Cy3 dyes were purchased from IDT and used directly as FISH probes.

## Antibodies

Antibodies against Ppa-RAD-51 were generated against a 6xHis-tagged N-terminal fusion protein (aa 1–103) purified from *E. coli*. Four mice were immunized with the antigen. Serum from one animal, designated S148, was used without purification at 1:300 dilution (Pocono Rabbit Farm and Laboratory, Canadensis, PA). Antibodies against Ppa-HOP-1 were generated by genetic immunization against aa 177–276 (SDIX, Newark, DE) and used in the following experiments at 1:300 dilution. Additional antibodies were purchased from commercial sources and diluted as follows: mouse anti-FLAG (1:500, Sigma #F1804), mouse anti-V5 (1:500, Thermo Fisher #R960-25), rabbit anti-V5 (1:250, Millipore Sigma #V8137), goat anti-HA (1:500, Novus Biologicals #NB600-362), and mouse anti α-tubulin, and clone DM1A (1:400, Millipore #05-829). Secondary antibodies raised in donkey and labeled with Alexa 488, Cy3, or Cy5 were used at 1:400 dilution (Jackson ImmunoResearch Laboratories).

## Cytological analysis

To stage animals for each experiment, 30–40 J4 larvae were picked from a PCR-verified heterozygous mother onto a fresh plate and allowed to develop for an additional 24 or 48 hr at 20 °C. Immunofluorescence and FISH methods for germline tissue were adapted from *C. elegans* (*Phillips et al., 2009a*), with minor modifications. Briefly, adult hermaphrodites were dissected on a clean coverslip in egg buffer containing 0.05% tetramisole and 0.1% Tween-20. Samples were fixed for 2 min in egg buffer containing 1% formaldehyde and then transferred to a 1.5 ml tube containing PBST. After 5 min, the buffer was replaced with ice-cold methanol and incubated at room temperature for an additional 5 min. Worms were washed twice with PBST, blocked with Roche blocking reagent, and stained with primary antibodies diluted in Roche blocking solution for 1.5–2 hr at room temperature. Samples were washed with PBST and incubated with secondary antibodies raised in donkeys and conjugated with Alexa-488, Cy3 or Cy5 (Jackson ImmunoResearch Laboratories, West Grove, PA). Worms were then incubated with 1 µg/ml DAPI in PBST, washed with PBST, and mounted in ProLong Gold (Invitrogen) before imaging.

For immunofluorescence in embryos, mixed-stage worms were washed off of 20 plates in distilled water, pelleted, and treated with household bleach diluted eightfold with distilled water for 5 min at room temperature. Embryos were collected by centrifugation and washed twice with PBS. To dissolve the vitelline membrane, a solution containing 2.6 ml of n-heptane, 2 ml of PBS, and 1% paraformaldehyde (final) was added to the embryo pellet for 5 min at room temperature with shaking. Treated embryos were collected by centrifugation, washed twice with 5 ml MeOH, three times with PBS, and incubated with Roche blocking reagent, primary, and secondary antibodies as described above. Embryos were mounted on agarose pads for imaging.

To label replicating DNA in proliferating germline stem cells, gonads of age-matched hermaphrodites were injected with 0.3 nM Cy3-dUTP solution. After 30 min of recovery on seeded plates, worms were dissected and fixed as for immunofluorescence, stained with DAPI, and mounted for imaging.

For FISH experiments, age-matched animals were dissected and fixed as for immunofluorescence experiments described above, except that the initial fixation was for 4 min in 2% formaldehyde. After incubation in ice-cold methanol, worms were washed with 2 × SSCT twice and

incubated in 50% 2× SSCT/50% formamide solution overnight at 37 °C. The next day, the worms were transferred to a small PCR tube, excess solution was removed from the sample, and 40 μl of hybridization mix containing 250 ng of each probe in hybridization buffer (3.5× SSC, 58% forma-mide, 12.75% dextran sulfate) was added. DNA was denatured by incubation in a thermocycler at 91 °C for 2 min and incubated overnight at 37 °C. Following overnight hybridization, samples were transferred to a 1.5 ml tube and washed twice with 2× SSCT. After 5 min, the solution was replaced with fresh 2× SSCT and mounted with ProLong Diamond Antifade Mountant with DAPI (Invitrogen).

Images were acquired as stacks of optical sections at 0.2 μm z-spacing using a DeltaVision Elite microscope (GE) with an Olympus 100 × NA 1.4 objective. All data were deconvolved using the constrained iterative algorithm included with the softWoRx package (GE) using default cycle numbers and other settings, and maximum-intensity projections were generated for most images presented here. Distances between FISH probes were calculated using 3D peak finding and distance calculation functions within softWoRx. Images were pseudocolored using softWoRx, Fiji (*Schindelin et al., 2012*), or Adobe Photoshop.

## Genetic map construction

We generated a genetic map of *P. pacificus* using a double-cross hybrid strategy that allowed the simultaneous mapping of meiotic COs in male spermatocytes and hermaphrodite oocytes (*Figure 8A*). We began with three parental lines (P0): 'CA,' the 'wild type' reference strain PS312 from California; 'WA,' PS1843 from Washington State; and 'LR,' RSB001 from La Réunion Island. These three strains have been previously used to generate genetic linkage maps (*Rödelsperger et al., 2017*; *Srinivasan et al., 2003*; *Srinivasan et al., 2002*). We crossed these P0 lines to obtain F1 hybrid CA/WA males and CA/LR hermaphrodites. These F1 hybrid types were then mated to produce the F2 generation, so that each F2 inherits a recombinant genome from both of its F1 parents.

F2 progeny were genotyped using a custom sequencing approach: juvenile (unmated) F2 hermaph-rodites were placed on individual 60 mm plates and allowed to self-fertilize for 2–3 generations until their bacterial food was depleted. Worms were washed from each plates and genomic DNA was extracted using Qiagen DNeasy Blood & Tissue Kit (Qiagen) and quantified by Qubit (Invitrogen). Because self- and cross-progeny can be temporally intermingled among broods of *P. pacificus*, we genotyped each sample using several SSLP markers (*Dieterich et al., 2008*) to confirm the presence of the PS1843 (WA) haplotype (*Supplementary file 3*). A total of 93 validated F2 samples were used to construct genomic libraries for whole-genome sequencing using the plexWell LP384 Library Prepa-ration Kit (seqWell), with ~5× shotgun sequence per F2 sample.

To obtain a set of reference variants for use in genotyping, we sequenced the three parental lines: CA and LR at 86× depth and WA at 61×, and mapped to the CA 'El Paco' reference genome using bwa-mem2 (v2.0pre2-4-gf06fc0c; *Md et al., 2019*). Using freebayes (v1.3.1–17-gaa2ace8; *Garrison and Marth, 2012*), with the `--standard_filters` option, we identified 1,395,065 polymorphic sites with that were (1) homozygous for each P0 genotype and within three standard deviations of the mean depth for all three (CA, 47–125, LR, 44–125, WA, 20–111) and (2) biallelic, with a different nucleotide in one strain relative to the other two. There were 374,862 such cross-informative sites where CA differed from both WA and LR; 490,484 where LR differed from CA and WA; and 529,719 where WA differed from CA and LR. Importantly, these sites were distributed across the genome.

We genotyped each F2 sample for these cross-informative variants in non-overlapping 100 kb blocks spanning the genome. Our crossing scheme gives rise to four possible genotypes at each position (CA/CA, CA/LR, CA/WA, and WA/LR) that can be easily differentiated based on the relative total depth $f_{CA}$, $f_{WA}$, and $f_{LR}$ at cross-informative variant sites within a block, where $f_{CA} + f_{WA} + f_{LR} = 1$. Specifically, we called CA/CA if $f_{CA} > 0.95$, CA/LR if $f_{WA} < 0.05$ and $|f_{CA} - f_{LR}| > 0.1$, etc. (*Figure 8—figure supplement 1A*). Genotypes not satisfying one of these inequalities were not called. Genomic blocks that contained fewer than 100 diagnostic sites or for which more than 30% of F2 individuals could not be called were not genotyped. Of the 1567 non-overlapping 100 kb blocks spanning the genome, 1398 remained after filtering for too few sites, too few individuals genotyped, and segregation distor-tion (p≤0.001).

By design, our cross generates sex-specific genetic maps since WA haplotypes can only be trans-mitted via the sperm of the CA/WA male parent and LR haplotypes can only be transmitted to ova of

the CA/LR hermaphrodite parent. Genetic linkage maps were constructed with OneMap (*Margarido et al., 2007*) using the 'F2 backcross' population type and the 'record' algorithm to order markers.

All sequencing data have been submitted to the NIH Sequence Read Archive under accession number PRJNA734516.

### Orthology analysis and phylogenetic inference

Accessions to all data used in the orthology analysis and phylogenetic inference are available in *Supplementary file 2*. We downloaded the genomes and annotation files for 66 nematode species (selecting a maximum of two species per genus) from WormBase ParaSite (*Howe et al., 2017*) and two tardigrade outgroup taxa (*Hysibius dujardini* and *Ramazzottius varieornatus*) from tardigrades. org. We selected the longest isoform of each protein-coding gene in each species using AGAT (version 0.5.1; *Dainat et al., 2020*). To infer the nematode phylogeny, we ran BUSCO (version 5.0.0; using the 'metazoa_odb10' dataset; *Simão et al., 2015*) on all filtered proteomes and used busco-2fasta.py (available at https://github.com/lstevens17/busco2fasta; *Stevens, 2021*, *Camacho et al., 2009*) to extract 534 single-copy orthologs that were present in at least 70% of species. We aligned the orthologous sequences using MAFFT (version 7.475; *Katoh and Standley, 2013*) and inferred gene trees using IQ-TREE (version 2.1.2; *Nguyen et al., 2015*), allowing the substitution model to be automatically selected (*Kalyaanamoorthy et al., 2017*). We provided the resulting gene trees to ASTRAL-III (version 5.7.4; *Zhang et al., 2018*) to infer a species tree. The resulting tree was visualized using iTOL (*Letunic and Bork, 2016*).

To investigate the presence of meiotic pairing proteins across the nematode phylogeny, we used OrthoFinder (version 2.5.2; *Emms and Kelly, 2015*) to cluster the longest isoforms of each protein-coding gene into orthologous groups (OGs). We identified the *P. pacificus* orthologs of DMC-1, MND-1, HOP-2, and RAD-51 by searching the proteome using BLASTP (version 2.5.0+); with the orthologous protein sequences from *Homo sapiens* as queries and identified the corresponding OGs. We identified the OG containing the HIM-8/ZIM family using the *C. elegans* transcript IDs. We aligned each OG using MAFFT and inferred a gene tree using IQ-TREE as previously described. Each gene tree was visually inspected using iTOL and presence or absence of a homolog was recorded for each species. For those species that lacked an apparent MND-1 or HOP-2 ortholog, we searched the genome sequence using TBLASTN with the *P. pacificus* orthologs as a query (recording evidence of a homolog as one or more hits with an E-value of <1e-5). For species that lacked a DMC-1 ortholog, we searched the genome sequence using both the *P. pacificus* DMC-1 and RAD-51 protein sequences as query, and considered hits (E-value <1e-5) for DMC-1 that were non-overlapping with those for RAD-51 as evidence of a DMC-1 ortholog. For species that lacked a member of the HIM-8/ZIM family, we searched the genome sequence using TBLASTN with the homologous sequence from its closest relative as a query.

## Acknowledgements

This work was supported by funding from the Howard Hughes Medical Institute and the Miller Institute to AFD, as well as a fellowship from the Helen Hay Whitney Foundation to JB. This work used the Vincent J Coates Genomics Sequencing Laboratory at UC Berkeley, supported by NIH S10 OD018174 Instrumentation Grant. We are grateful to Ralf J Sommer, Christian Rödelsperger, Ray Hong, and other members of the *P. pacificus* research community for providing strains and abundant helpful advice. We thank members of the Dernburg Lab for many discussions about this work and critical reading of the manuscript. We are grateful to the thoughtful reviewers of the original version of our manuscript, particularly for spurring us to produce a genetic map.

## Additional information

### Funding

| Funder | Grant reference number | Author |
|--------|------------------------|--------|
| Howard Hughes Medical Institute | | Abby F Dernburg |
| Miller Institute for Basic Research in Science | | Abby F Dernburg |
| Helen Hay Whitney Foundation | Fellowship | Joshua Bayes |

The funders had no role in study design, data collection and interpretation, or the decision to submit the work for publication.

### Author contributions

Regina Rillo-Bohn, Conceptualization, Formal analysis, Investigation, Supervision, Writing – original draft, Writing – review and editing; Renzo Adilardi, Formal analysis, Investigation, Methodology, Writing – original draft, Writing – review and editing; Therese Mitros, Data curation, Formal analysis; Barış Avşaroğlu, Clara Wang, Sabrina Lin, K Alienor Baskevitch, Investigation; Lewis Stevens, Data curation, Formal analysis, Methodology; Simone Köhler, Investigation, Writing – review and editing; Joshua Bayes, Conceptualization, Investigation; Daniel S Rokhsar, Methodology, Supervision, Writing – review and editing; Abby F Dernburg, Conceptualization, Funding acquisition, Investigation, Methodology, Project administration, Supervision, Writing – review and editing

### Author ORCIDs

Renzo Adilardi (iD) http://orcid.org/0000-0001-7279-3853
Lewis Stevens (iD) http://orcid.org/0000-0002-6075-8273
Abby F Dernburg (iD) http://orcid.org/0000-0001-8037-1079

### Decision letter and Author response

Decision letter https://doi.org/10.7554/eLife.70990.sa1
Author response https://doi.org/10.7554/eLife.70990.sa2

## Additional files

### Supplementary files

• Supplementary file 1. Table listing genes, alleles, and genome editing reagents used in this manuscript.

• Supplementary file 2. Table listing species names and sequence accession numbers used for the phylogenetic analysis summarized in *Figure 2—figure supplement 1*.

• Supplementary file 3. Table listing SSLP markers and genotyping primers used to confirm cross-progeny for genetic map construction.

• Supplementary file 4. Table listing software version numbers and parameters used in this work.

• Supplementary file 5. Table showing summary of crossovers detected on each chromosome in hermaphrodite oocytes (XX) and male spermatocytes (XO), including the number of nonrecombinant chromatids and chromatids with one or two crossovers. No higher-order recombinant chromatids were observed.

• Transparent reporting form

### Data availability

Sequence data used to generate the recombination map for P. pacificus have been deposited at the NIH Sequence Read Archive under accession number PRJNA734516 and are available at https://www.ncbi.nlm.nih.gov/bioproject/PRJNA734516. These include sequence reads for three parental genomes and 93 hybrid progeny. Genotype calls are provided as Excel files in the Supplemental Data.

The following dataset was generated:

| Author(s) | Year | Dataset title | Dataset URL | Database and Identifier |
|---|---|---|---|---|
| Adilardi R, Mitros T, Rokhsar DS, Dernburg AF | 2021 | Mapping meiotic crossovers in Pristionchus pacificus hybrids | https://www.ncbi.nlm.nih.gov/bioproject/PRJNA734516 | NCBI BioProject, PRJNA734516 |

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
