## [Decision Letter]

**Acceptance summary:**

Meiosis, and specifically the events that ensure proper chromosome segregation at meiosis I, have been mostly explored in model organisms. This study provides a novel view by exploring these processes in a nematode, *Pristionchus pacificus*, distantly related to the model nematode species *C. elegans*. The authors identified several differences in the control of homolog pairing between these nematodes. This work highlights the importance of exploring the evolutionary diversity of meiosis.

**Decision letter after peer review:**

Thank you for submitting your article "Analysis of meiosis in *Pristionchus pacificus* reveals plasticity in homolog pairing and synapsis in the nematode lineage" for consideration by *eLife*. Your article has been reviewed by 3 peer reviewers, including Bernard de Massy as the Reviewing Editor and Reviewer #1, and the evaluation has been overseen by a Reviewing Editor and Jessica Tyler as the Senior Editor.

The reviewers have discussed the reviews with one another and the Reviewing Editor has drafted this decision to help you prepare a revised submission.

Summary:

The manuscript by Rillo-Bohn et al. reports the cytological and functional analysis of meiotic prophase in a nematode Pristionchus pacificus, who diverged 200-300 million years ago from *C. elegans* but sharing most biological properties and genomic features.

Unlike *C. elegans*, the *P. pacificus* genome contains clear orthologs of DMC-1 (and MND-1) but not pairing center proteins (ZIM/HIM), critical for meiotic chromosome pairing/synapsis in *C. elegans*. In *P. pacificus*, pairing and synapsis depend on Spo-11 and Dmc-1 but not Rad-51. Dmc-1 is required for bivalent formation, not Rad-51.

Surprisingly, RAD-51 is not important for crossing over in *P. pacificus*. Further, the authors show that DMC-1 and RAD-51 have spatio-temporally distinct localization patterns and are not dependent on each other for localization,

Based on cytological analyses and previously published genetic maps, the authors conclude that crossovers result from both class I and II crossover pathways.

Essential revisions:

Overall, the paper is quite interesting and is adding important and novel information for understanding features of meiosis and their evolution. In general, the data needs to be described with more accuracy and conclusions presented in the text should be supported by the data with quantification (see below). Some conclusions presented in the discussions are over-interpretation of the data and should be revised.

Two conclusions presented in this paper must be clarified as they bring unexpected views about meiotic prophase:

1) Spo-11 dependent DMC-1 foci not associated to recombination intermediates.

2) Class II crossovers without chiasma (but check genetic map data, see below).

1) Whether DMC-1 foci correspond to recombination or not should be determined.

Several issues emerge from the data as it is presented : Quantify DMC-1 foci (this also applies to RAD-51). Line 292 indicates DMC-1 coats the chromosomes. This data needs to be clarified. Presence of such DMC-1 localization would be extremely puzzling and in contradiction with all current cytogical data on DMC1 localization in many species.

Either use higher resolution microscopy and/or test another tag, as the V5 tag may create artefacts, despite being functional. Localization of DMC-1 with respect to asynapsed/synapsis should be determined. Conclusions should be based on quantitative analysis not only by showing a few nuclei. As mentioned above, on Line 321, the authors conclude that DMC-1 locates broadly on chromatin rather than specifically to recombination intermediates. This statement needs to be supported by the data.

Line 298. It is concluded that DMC-1 and RAD-51 do not overlap (Figure 4C). But the legend indicates that the foci do not completely overlap. Please clarify.

Line 330. If the authors' interpretation is that these RAD-51 foci result from endogenous breaks, it should be mentioned.

Meiosis II should be analyzed in rad-51 and dmc-1 mutants? Are there broken chromatids? For instance, if there are broken chromatids in rad-51 mutants not in dmc-1, this could suggest that Rad-51 mediates the repair of DSB between sister chromatids.

2) Class II crossovers. The authors should verify information about the genetic map :

L456: Where does the information of 100-250 cM per chromosome come from? This statement does not fit the published genetic maps (Srnivasan et al., 2002, 2003) where genetic maps are between 30 and 100cM per chromosome for a total of 338cM for the 6 linkage groups (although the 2003 paper seems to indicate higher values). It does not fit either the more recent genetic map, about 310cM total (Rodelsperger et al., 2017). It seems that most chromosomes have 1 CO which would then fit the bivalent data. In addition, those maps show a strongly reduced CO rate in the center of chromosomes, potentially a similarity with *C. elegans* CO distribution?

If the conclusion is that there should be class II crossovers, the authors should validate this conclusion by analyzing cosa-1 and mus-81/eme-1 mutants. They should also provide data and interpretation for the presence of only one chiasma per bivalent (ie the statement about cruciform bivalent in Line 293).

Figure 1, the authors show Cy3-dUTP labeling in the *P. pacificus* germline. There was no mention of this experiment in the Results section. We recommend either removing it, or specifically reporting/discussing this finding. Similarly, in Materials and methods, RNAseq is included but we see no place where it is used.

Figure 1C, several points should be clarified : indicate the meiotic stages of those nuclei. Is the embryo nuclei at metaphase ? Are there chromatin bridges in the nucleus in the middle?

Figure 2, Analysis of pairing.

It is proposed that decreased distances in spo-11 and dmc-1 mutants are due to telomere clustering. One needs to know what is the frequency of clustering-like nuclei in TZ 2 to 4 (wt and mutants), and in which zones were analyzed for the heterologous controls (2E).

Figure 2D: Presumably, the horizontal bars show median and range of the middle two quartiles, but this is never defined. It is very difficult to see these bars in many of the charts, either because the bars are behind and masked by the data points or because the bars are yellow. Mean and SD should be given, either in the figure or in the text.

Figure 2E legend: Indicate that heterologous signals are between X and IV.

Figure 2 supplement 1.

Clade I is not labelled, please revise ; Clade II and IV have no ortholog for the four proteins tested ? Is this due to issues in genome assemblies ? Do they have Rad-51 orthologs ? Two columns should be added : one for Rad-51 and one for Spo-11.

Figure 3: There appears to be one to two tracks of SYP-4 in almost all of the spo-11 and dmc-1 mid-prophase nuclei, suggesting that there is some synapsis in the absence of these proteins. Could this "synapsis" be non-homologous? Could it be the X?

Figure 5B: the text and figure only show the mean. At a minimum, they should also include the SD.

Figure 5D: It is very difficult to understand the staining shown here. The region of the spermatheca should be indicated. The six bivalents cannot be seen nuclei with, even in WT. Symbols need to be included in the figure to highlight the important points.

Figure 5E: for % Egg viability, what was the number of eggs laid that were counted? Is this different than the value in the "Eggs laid" column? For "% Male progeny," what was the number of adult worms counted? This also applies to Figure 3 supplement 1.

Figure 7: What are the residuals COSA-1 foci in spo-11 mutant?

Line 22: last eukaryotic common ancestor (LECA) is only used once, so there is no need to define an abbreviation.

Line 42: Revise citation: two papers led to the conclusion that meiotic recombination was initiated by a conserved topoisomerase-like enzyme called Spo11: Bergerat et al., 1997 and Keeney et al., 1997.

Line 113: The switch from spermatogenesis to oogenesis occurs in early adulthood in *P. pacificus*. This seems to be a difference with *C. elegans*. Please clarify.

Line 118. Please highlight the parallel DAPI tracks, they are not visible on Figure 1A.

Line 158: Clade V should be defined as the clade that includes both *C. elegans* and *P. pacificus*.

Line 166: co-CRISPR has "not been helpful for *P. pristionchus*." What does this mean?

Line 244, refer to Figure 3.

Line 267: The use of « pachy-diplotene » should be clarified. If the nuclei are only partially synapsed chromosomes, then these are diplotene ; alternatively it may refer to a zone where there is juxtaposition of pachytene and diplotene nuclei. But then what does early, mid and late mean is not clear.

Line 301: typo – typical.

Line 311: "RAD-51 was also observed in some nuclei in dmc-1 mutants" this wording suggests fewer nuclei than normal have RAD-51 foci, but the figure makes it look like most nuclei have RAD-51 foci once they escape from the extended transition zone. Please quantify.

Line 384: typo.

Lines 393-395: Provide data showing that all bivalents have a cruciform structure.

Lines 397-399: Quantification is needed to support the claim of 0 and 6 COSA-1 foci in dmc-1 and rad-51 mutants. At the very least, this needs to be given as a mean +/- SD.

In the discussion, please revise interpretations:

Lines 428-430: What is the evidence that association of RAD-51 with recombination intermediates is only permitted after DMC-1 has ensured the formation of CO intermediates? RAD-51 forms foci in dmc-1 mutants.

Lines 434-436: What is the data suggesting that RAD-51 processes "excess" DSBs that remain after CO designation? This would suggest that RAD-51 is not involved in processing DSBs into CO DSBs, there does not seem to be data to substantiate this claim.

Table S1: Revise Hop-1 raw, information missing.

Authors should mention that SYP-4::HA staining does not necessarily imply that the tripartite SC is formed.

Revise use of italics throughout the text : ie Line 11, 13, 87.

J4- Please define this stage for non-experts.

Define COSA-1 when first used.

Class I and Class II crossovers are first defined in the Discussion section, but they are repeatedly referred to in the introduction and results. An earlier definition seems appropriate.

For all genetic modifications (null mutations and tag insertion), indicate the protein change (truncation, deletion, position of tag…). Indicate whether epitope-tagged proteins are heterozygous or homozygous.

Respect order of figure presentation: figure 5 cannot come in L178.

[Editors' note: further revisions were suggested prior to acceptance, as described below.]

Thank you for submitting your article "Analysis of meiosis in *Pristionchus pacificus* reveals plasticity in homolog pairing and synapsis in the nematode lineage" for consideration by *eLife*. Your article has been reviewed by 3 peer reviewers, including Bernard de Massy as the Reviewing Editor and Reviewer #1, and the evaluation has been overseen by a Reviewing Editor and Jessica Tyler as the Senior Editor.

This study reports an important analysis of meiosis in *Pristionchus pacificus*, a nematode distantly related to *C. elegans*. The analysis is based on developing many tools, characterizing homologs of key meiotic proteins such as to perform a cytological and genetic analysis of meiosis.

The authors show this species has orthologs of SPO11, DMC1, RAD51, HOP1, SYP-4, COSA-1.

They show that pairing and synapsis depend on SPO11 and DMC1, not on RAD51.

Major insights are:

The differential function of DMC1 and RAD51, DMC but not RAD51 being required for pairing and synapsis.

The differential localization of DMC1 and RAD51, mostly during TZ for DMC1 and later stages for RAD51.

The SPO11 dependent synapsis.

CO control with one CO and one COSA-1 focus per bivalent, and oriented desynapsis.

A genetic map coherent with the cytology.

The paper is clearly written, and the data and interpretation are convincing.

Essential revisions:

1) Ideally to validate the DMC1 staining, the authors would need to exclude the possibility that it is not due to the V5 tag. The fact that RAD51::V5 behaves as RAD51 does not allow to conclude on the potential effect of the V5 tag on DMC1. Otherwise, the authors should mention explicitly the possibility that the broad DMC1 staining observed could be a specificity of the tagged protein.

2) Data about orthologs and protein alignment should be provided (SPO11, DMC1, HOP2, MND1, RAD51, HOP1, SYP4, COSA1).

3) RAD-51 foci in wild type and Dmc1 mutant should be quantified to determine whether there is a detectable and significant difference.

L41. Is.

L122. Please explain this apparently synchronyzed group of cells in S phase.

Lines 129-131, Figure 1A: the text sets up that the region of decondensation between diplotene and diakinesis has been called the "growth zone" in Ppa and the "diffuse stage" in other organisms, but the figure calls this region in Ppa "Diff" for diffuse stage. I am ok with this nomenclature, but the authors may want to say in the text or legend that for the sake of consistency with other organisms, they will call this the diffuse stage.

L142. Figure 1D. Please clarify the staining in this panel D: one expects 12 DAPI staining bodies, or 12 CENP-C structures ? More than 12 are visible?

L179. What do you mean by "frequent".

L181. What is the plate phenotype?

L189. Figures should be numbered following the text.

Lines 200-216: For some reason, I got confused when I read, "in wild-type *C. elegans*" in line 212 and wasn't sure what genotype was described earlier. This was probably just due to a tired brain, but it might help the reader if line 204 began, "In zone 1 of wild-type *P. pacificus* hermaphrodites," and then "in wild-type animals" could be deleted from line 207. This way, the reader would have the genotype in their mind before seeing the data.

L218. Please clarify: in wild type, the pairing is decreased in the zones following meiotic entry, ie after the transition zone.

L220. Rad51 mutant seems to have a phenotype in zone 2?

Please quantify heterologous distances in Rad51 mutant.

Lines 274-287, Figure 3B: The nuclei labeled PM have puncta of SYP-4 and appear to have weak HOP-1 associated with chromosomes rather than excluded from chromosomes, which one might expect if the expression were truly nucleoplasmic. Is it possible that these nuclei are early leptotene, and that in *P. pacificus* adoption of the crescent shape is not concomitant with the onset of meiosis? It is curious that in the full gonad image in 3A, more distal nuclei do not appear to express SYP-4 or HOP-1, which makes me wonder if the "PM" nuclei are truly premeiotic or very early meiotic.

L288. The authors should add a comment about the SYP-4 lines in Spo11-1 mutant (as indicated in the rebuttal letter)(non homologous? not the X?).

Line 305, etc: It's clearly beyond the scope of this manuscript, but it would be really interesting to look at DMC-1 localization in *P. pacificus* spo-11 mutants irradiated with a low dose of γ irradiation to see if at very low DSB number DMC-1 "coats" the chromosomes or forms puncta, and whether there is a DSB number or density at which staining switches to the broad distribution seen in wt.

L338. Not necessarily dsDNA, DMC1 could interact with proteins.

L351: It seems that two alternatives are: two rounds of DSBs? or two rounds of DSB repair?

L381. Please comment on why egg viability is reduced.

L468. How are the two arms defined?

L484 and 585. Why ? I understood the low CO activity is due to an inversion or other rearrangement, why then proposing a PC effect?

L536. This is an over-simplification, the mouse data is based on X ray irradiations.

Line 541: it is interesting that *C. elegans* rad-51 mutants have fragmented and "clumpy" chromosomes, and these are not seen in either Ppa dmc-1 or rad-51 mutants. Have you looked at a double mutant? I know this would be a tricky experiment without balancers (unless dmc-1 and rad-51 happen to be on the same chromosome).

L991. This sentence does not belong to panel A.

*Reviewer #1 (Recommendations for the authors):*

The authors developed multiple tools to investigate meiosis in Pristionchus pacificus, a nematod distantly related to *C. elegans*.

This required a number of technological developments suitable for this species:

CrispR/Cas9 editing

Cytology

Raising antibodies or adding tags to selected proteins

Generating a genetic map

Evaluating protein conservation by Blast or other alignment tools

The authors show the presence of both DMC1 and RAD51, but only DMC1 being required for meiosis. This is a novel and interesting separation of functions between these two proteins.

Also, it is shown that synapsis and homolog pairing is dependent on DSB formation in *P. pacificus*, unlike *C. elegans*.

Based on the phylogeny, it thus appears that *C. elegans* has lost DMC1 and gained a DSB independent pathway for homolog pairing.

This study adds an important novel view on the diversity of meiosis, and specifically highlights a regulation of DSB repair by DMC1 and RAD51 that will be interesting to unravel.

This study reports an important analysis of meiosis in Pristionchus pacificus, a nematode distantly related to *C. elegans*. The analysis is based on developing many tools, characterizing homologs of key meiotic proteins such as to perform a cytological and genetic analysis of meiosis.

The authors show this species has orthologs of SPO11, DMC1, RAD51, HOP1, SYP-4, COSA-1

They show that pairing and synapsis depend on SPO11 and DMC1, not on RAD51.

The differential function of DMC1 and RAD51

The differential localization of DMC1 and RAD51, mostly during TZ for DMC1 and later stages for RAD51.

The SPO11 dependent synapsis

CO control with one CO and one COSA-1 focus per bivalent, and oriented desynapsis.

A genetic map coherent with the cytology.

The paper is clearly written, and the data and interpretation are convincing

1) to validate the DMC1 staining to determine whether the broad and intense staining indicated to extend to other sites than recombination sites is an artifact of the DMC1-V5 fusion or not.

2) Data about orthologs and protein alignment should be provided (SPO11, DMC1, HOP2, MND1, RAD51, HOP1, SYP4, COSA1).

3) RAD-51 foci in wild type and Dmc1 mutant should be quantified to determine whether there is a detectable and significant difference.

L41. Is.

L122. Please explain this apparently synchronyzed group of cells in S phase.

L142. Figure 1D. Please clarify the staining in this panel D: one expects 12 DAPI staining bodies, or 12 CENP-C structures ? More than 12 are visible?

L179. What do you mean by "frequent".

L181. What is the plate phenotype?

L189. Figures should be numbered following the text.

L218. Please clarify: in wild type, the pairing is decreased in the zones following meiotic entry, ie after the transition zone.

L220. Rad51 mutant seems to have a phenotype in zone 2?

Please quantify heterologous distances in Rad51 mutant.

L338. Not necessarily dsDNA, DMC1 could interact with proteins.

L351: It seems that two alternatives are: two rounds of DSBs? or two rounds of DSB repair?

L381. Please comment on why egg viability is reduced.

L468. How are the two arms defined?

L484 and 585. Why? I understood the low CO activity is due to an inversion or other rearrangement, why then proposing a PC effect?

L536. This is an over-simplification, the mouse data is based on X ray irradiations.

L991. This sentence does not belong to panel A.

*Reviewer #2 (Recommendations for the authors):*

The revised manuscript by Rillo-Bohn et al. has largely addressed the initial reviews. This has included additional experiments, most notably reexamining the genetic map, and being more precise in the description, analysis and conclusion of the experiments. One point that does not appear to be addressed (or if was, I missed it), concerns the single tracks of SYP-4 observed in the spo-11 and dmc-1 mutants (Figure 3). While the authors indicated in the response to reviewers that "Based on our pairing analysis, the synapsis must be nonhomologous and does not appear to occur preferentially on the X chromosome. We have added a comment to this effect. We have seen other examples in other mutants (beyond the scope of this paper)".

I do not see any mention of this in the manuscript. Lines 288-290. It also might be worth mentioning that similar tracks have been observed in *C. elegans*.

Typos:

Line 41-42: missing "is".

Line 252: the reference is formatted funny.

*Reviewer #3 (Recommendations for the authors):*

The manuscript by Dr. Abby Dernburg and colleagues establishes the nematode *Pristionchus pacificus* as a model for studies of meiosis and demonstrates that unlike the more well studied nematode *Caenorhabditis elegans*, Pristionchus retains the meiosis-specific recombinase Dmc1 and depends on recombination intermediates for stable homolog pairing and synapsis. Thus, in this regard pairing and synapsis during Pristionchus meiosis more closely resemble that of budding yeast and mammals than that of *C. elegans*, which lacks Dmc1 and instead relies on pairing centers rather than recombination for pairing and synapsis.

This revised manuscript not only addresses the majority of reviewer comments, it goes for beyond the requests to reviewers, for example, adding a newly derived genetic map that clarifies the recombination landscape in Pristionchus and demonstrates cases of transmission distortion and hybrid incompatibilities. These newly added findings will be of great interest to researchers using this nematode and to evolutionary biologists. Thus, the changes in this version not only strengthen the conclusions, they enhance the overall impact of the paper.

Also of great interest are several unexpected findings that are described in the manuscript and will undoubtedly be the basis of future mechanistic studies. The broad localization of Ppa-DMC-1 on meiotic chromosomes rather than to single-stranded 3' overhangs specifically at DSB sites as expected from studies of other organisms is surprising, and may have implications for how DMC-1 mediates strand invasion. It is also curious that, although Ppa-DMC-1 and Ppa-RAD-51 both associate with meiotic chromosomes, RAD-51 does not function as an accessory protein that promotes DMC-1 filament assembly as has been described in other organisms that utilize both paralogues. Finally, given that reliance on recombination-independent mechanisms for pairing and synapsis correlates with absence of Dmc1 and its accessory factors in *Drosophila* and Caenorhabditis, it is very interesting that the genomes of most Clade V nematodes encode for Dmc1 orthologs and pairing center proteins, and even Pristionchus may retain some pairing center-like activities.

In summary, this paper is notable for well-conducted and described experiments that define pairing and synapsis mechanisms in Pristionchus as well as unexpected findings that do not presently lead to a mechanistic understanding but clearly whet the appetite for future studies in Pristionchus.

---

## [Author Response]

Essential revisions:Overall, the paper is quite interesting and is adding important and novel information for understanding features of meiosis and their evolution. In general, the data needs to be described with more accuracy and conclusions presented in the text should be supported by the data with quantification (see below). Some conclusions presented in the discussions are over-interpretation of the data and should be revised.Two conclusions presented in this paper must be clarified as they bring unexpected views about meiotic prophase:1) Spo-11 dependent DMC-1 foci not associated to recombination intermediates.2) Class II crossovers without chiasma (but check genetic map data, see below).1) Whether DMC-1 foci correspond to recombination or not should be determined.Several issues emerge from the data as it is presented : Quantify DMC-1 foci (this also applies to RAD-51). Line 292 indicates DMC-1 coats the chromosomes. This data needs to be clarified. Presence of such DMC-1 localization would be extremely puzzling and in contradiction with all current cytogical data on DMC1 localization in many species.Either use higher resolution microscopy and/or test another tag, as the V5 tag may create artefacts, despite being functional. Localization of DMC-1 with respect to asynapsed/synapsis should be determined. Conclusions should be based on quantitative analysis not only by showing a few nuclei. As mentioned above, on Line 321, the authors conclude that DMC-1 locates broadly on chromatin rather than specifically to recombination intermediates. This statement needs to be supported by the data.

We have now addressed this concern by tagging RAD-51 with the same tag (V5), which resulted in sparse, punctate staining essentially identical to that seen with our polyclonal antibody raised against RAD-51. This rules out the possibility that the more diffuse staining and differential timing seen for DMC-1 was a consequence of the localization method or antibodies. We also tried to tag DMC-1, but were unsuccessful after multiple attempts, and felt that it was not worth expending further effort on this in light of our findings with RAD-51::V5.

Line 298. It is concluded that DMC-1 and RAD-51 do not overlap (Figure 4C). But the legend indicates that the foci do not completely overlap. Please clarify.

We have revised the legend and text for clarity. These proteins are only occasionally detected in the same nucleus and when they are their localization does not overlap. This is consistent with our evidence from mutants that the two paralogs can function independently of each other.

Line 330. If the authors' interpretation is that these RAD-51 foci result from endogenous breaks, it should be mentioned.

The RAD-51 foci in the mitotic region of the gonad likely reflect replication-associated DNA damage.

Meiosis II should be analyzed in rad-51 and dmc-1 mutants? Are there broken chromatids? For instance, if there are broken chromatids in rad-51 mutants not in dmc-1, this could suggest that Rad-51 mediates the repair of DSB between sister chromatids.

This is not easy to do in nematodes since Meiosis II is quite transient. However the appearance of oocytes at diakinesis can reveal defects in DNA repair. For example, in *C. elegans rad-51* or *nbs^-1^* mutants, or in *mre-11* separation-of function mutants, the chromosomes at diakinesis are typically fragmented and “clumpy” (likely due to NHEJ) and we do not see this in either *dmc-1* or *rad-51* mutants of *P. pacificus*. Thus we conclude that neither DMC-1 nor RAD-51 is essential to repair meiotic DSBs.

2) Class II crossovers. The authors should verify information about the genetic map :L456: Where does the information of 100-250 cM per chromosome come from? This statement does not fit the published genetic maps (Srnivasan et al., 2002, 2003) where genetic maps are between 30 and 100cM per chromosome for a total of 338cM for the 6 linkage groups (although the 2003 paper seems to indicate higher values). It does not fit either the more recent genetic map, about 310cM total (Rodelsperger et al., 2017). It seems that most chromosomes have 1 CO which would then fit the bivalent data. In addition, those maps show a strongly reduced CO rate in the center of chromosomes, potentially a similarity with *C. elegans* CO distribution?If the conclusion is that there should be class II crossovers, the authors should validate this conclusion by analyzing cosa-1 and mus-81/eme-1 mutants. They should also provide data and interpretation for the presence of only one chiasma per bivalent (ie the statement about cruciform bivalent in Line 293)

We are grateful to the reviewer for pointing out the discrepancies between the earlier and more recent work from the Sommer lab, and for suggesting that this be reexamined. Although it has taken an inordinately long time (due in part to the pandemic) we have now constructed sex-specific recombination maps for *P. pacificus*, which have clearly shown that the early estimates of genetic distances were greatly inflated. Indeed, the map for *P. pacificus* is very similar to that for *C. elegans*, in that it shows elevated recombination on the arms (as noted by the reviewer) and a single crossover per homolog pair in most meiosis. The map also revealed evidence for crossover suppression and genetic incompatibilities between strains, both of which suggest directions for future studies.

Figure 1, the authors show Cy3-dUTP labeling in the P. pacificus germline. There was no mention of this experiment in the Results section. We recommend either removing it, or specifically reporting/discussing this finding. Similarly, in Materials and methods, RNAseq is included but we see no place where it is used.

We appreciate that these issues were called out and have now clarified both of these issues. Cy3-dUTP injection was used to pulse-label replicating chromosomes to confirm that the distal germline contains a region of proliferating germline stem cells, as in *C. elegans*. We have removed the reference to RNA-seq, which was used here only to help verify the structure and expression of meiotic genes, and will be presented in separate work.

Figure 1C, several points should be clarified : indicate the meiotic stages of those nuclei. Is the embryo nuclei at metaphase ? Are there chromatin bridges in the nucleus in the middle?

These images have now been updated with clearer examples and the figure legend has been edited for clarity. Extended kinetochore structures are most clearly visualized in embryonic nuclei at prometaphase, before they are fully condensed.

Figure 2, Analysis of pairing.It is proposed that decreased distances in spo-11 and dmc-1 mutants are due to telomere clustering. One needs to know what is the frequency of clustering-like nuclei in TZ 2 to 4 (wt and mutants), and in which zones were analyzed for the heterologous controls (2E).

The proximity between both homologous and heterologous FISH probes increases during the transient leptotene-zygotene stage, which we attribute to chromosome (not telomere) clustering. We have looked for telomere clustering using FISH and have not observed it; this makes it fairly subjective to define nuclei in which chromosomes are clustered. The absence of a discrete hallmark of clustered nuclei also makes it difficult to quantitatively correct for the differences in persistence of the clustered stage between wild-type and mutants. Despite these issues, we think that the conclusions of the pairing analysis are clear and consistent. We have revised our description of the pairing analysis to make our approach easier to understand.

Figure 2D: Presumably, the horizontal bars show median and range of the middle two quartiles, but this is never defined. It is very difficult to see these bars in many of the charts, either because the bars are behind and masked by the data points or because the bars are yellow. Mean and SD should be given, either in the figure or in the text.Figure 2E legend : Indicate that heterologous signals are between X and IV.

We have revised the graphs to avoid having the bars obscure the data points and the figure legend now states that the bars indicate the mean ± SD.

Figure 2 supplement 1.Clade I is not labelled, please revise; Clade II and IV have no ortholog for the four proteins tested ? Is this due to issues in genome assemblies ? Do they have Rad-51 orthologs? Two columns should be added : one for Rad-51 and one for Spo-11.

We elected to redo this analysis since additional nematode genomes have been sequenced and/or annotated since our original submission. The figure has been revised and all of these issues have been addressed except that we did not search for SPO-11. RAD-51 was found in all genomes, as mentioned in the text (but no longer included in the figure, for simplicity).

Figure 3: There appears to be one to two tracks of SYP-4 in almost all of the spo-11 and dmc-1 mid-prophase nuclei, suggesting that there is some synapsis in the absence of these proteins. Could this "synapsis" be non-homologous? Could it be the X ?

Based on our pairing analysis, the synapsis must be nonhomologous and does not appear to occur preferentially on the X chromosome. We have added a comment to this effect.

Figure 5B: the text and figure only show the mean. At a minimum, they should also include the SD.

The SD is not usually shown for this type of analysis since the major source of variance is the resolution of the microscope relative to the distance between DAPI-staining bodies, and is thus not biologically meaningful.

Figure 5D: It is very difficult to understand the staining shown here. The region of the spermatheca should be indicated. The six bivalents cannot be seen nuclei with, even in WT. Symbols need to be included in the figure to highlight the important points.

We removed these images since we agreed that they were unclear and also unnecessary.

Figure 5E: for % Egg viability, what was the number of eggs laid that were counted? Is this different than the value in the "Eggs laid" column? For "% Male progeny," what was the number of adult worms counted? This also applies to Figure 3 supplement 1.

We have updated these tables to include the number of broods analyzed (in the first column), mean number of eggs laid (± SD), and viability, expressed as a percent of eggs laid that reach adulthood.

Figure 7 : What are the residuals COSA-1 foci in spo-11 mutant?

2 aggregates per nucleus that are not associated with crossover intermediates (bivalents are never observed); we assume the same is true in *P. pacificus*, since we do not detect chiasmata in *spo-11* mutants.

Line 22: last eukaryotic common ancestor (LECA) is only used once, so there is no need to define an abbreviation.

True, but LECA is one of those acronyms that some people know better than the spelled-out version. We have now removed the abbreviation.

Line 42: Revise citation: two papers led to the conclusion that meiotic recombination was initiated by a conserved topoisomerase-like enzyme called Spo11: Bergerat et al., 1997 and Keeney et al., 1997.

We have often cited reviews rather than the primary sources for well-established information (e.g. Dernburg *et al.* 1998 was the first paper to demonstrate a conserved role for Spo11 in metazoan meiosis, but we did not cite that).

Nevertheless, we have revised this citation in light of the reviewer’s preference.

Line 113: The switch from spermatogenesis to oogenesis occurs in early adulthood in P. pacificus. This seems to be a difference with *C. elegans*. Please clarify.

The switch happens in early adulthood in both species, but the period of spermatogenesis extends for longer beyond the final molt in *P. pacificus*. We have deleted this statement to avoid confusion.

Line 118. Please highlight the parallel DAPI tracks, they are not visible on Figure 1A.

See also Figures 3, 4, and Figure 4 Supplement 1, where the parallel tracks of paired homologs are more evident.

Line 158: Clade V should be defined as the clade that includes both *C. elegans* and *P. pacificus*.

This has now been clarified on the figure and in the text.

Line 166: co-CRISPR has "not been helpful for P. pristionchus." What does this mean?

We have rephrased this for clarity. Many *C. elegans* researchers use “coCRISPR,” a technique in which worms are injected with a guide RNA and template to modify a gene with an easily-detected phenotype as a way to enrich identify successfully injected animals and enrich for edited progeny.

Line 244, refer to Figure 3.

Fixed.

Line 267 : The use of « pachy-diplotene » should be clarified. If the nuclei are only partially synapsed chromosomes, then these are diplotene ; alternatively it may refer to a zone where there is juxtaposition of pachytene and diplotene nuclei. But then what does early, mid and late mean is not clear.

We have now changed all mentions of “pachy-diplotene” to simply “diplotene” for simplicity, although we are unaware of examples outside of nematodes in which such an extended diplotene period is observed. The Figures and associated legends have also been updated.

Line 301: typo – typical.

Fixed.

Line 311: "RAD-51 was also observed in some nuclei in dmc-1 mutants" this wording suggests fewer nuclei than normal have RAD-51 foci, but the figure makes it look like most nuclei have RAD-51 foci once they escape from the extended transition zone. Please quantify.Line 384: typo.

Fixed.

Lines 393-395: Provide data showing that all bivalents have a cruciform structure.

We have now added a new image (Figure 1C) that displays the bivalent structure more clearly. In light of our new evidence for a single CO per chromosome, it no longer seems controversial to argue that most bivalents have a single chiasma.

Lines 397-399: Quantification is needed to support the claim of 0 and 6 COSA-1 foci in dmc-1 and rad-51 mutants. At the very least, this needs to be given as a mean +/- SD.

We do not feel that quantification would be helpful here; the images clearly show that COSA-1 forms normal numbers of foci in *rad-51* mutants but are greatly reduced or absent in *dmc-*1 mutants.

In the discussion, please revise interpretations:Lines 428-430: What is the evidence that association of RAD-51 with recombination intermediates is only permitted after DMC-1 has ensured the formation of CO intermediates? RAD-51 forms foci in dmc-1 mutants.

We have rephrased this for clarity. Based on the extended transition zone observed in *dmc-1* mutants, as well as a body of findings that the extended transition zone in *C. elegans* reflects the activity of meiotic checkpoint mechanisms, it seems likely that RAD-51 normally localizes to nuclei following satisfaction of the obligate crossover checkpoint. However, in *dmc-1* mutants nuclei eventually progress to a stage that is permissive for RAD-51-mediated repair although they lack designated COs.

Lines 434-436: What is the data suggesting that RAD-51 processes "excess" DSBs that remain after CO designation? This would suggest that RAD-51 is not involved in processing DSBs into CO DSBs, there does not seem to be data to substantiate this claim.

In light of our new genetic evidence for a single crossover per chromosome pair, as well as the absence of univalent chromosomes in rad-51 mutants, we can conclude that RAD-51 is not involved in processing DSBs into COs.

Table S1: Revise Hop-1 raw, information missing.

We did not include any strains in which *hop-1* was epitope-tagged or mutated; we included it in the table just to indicate the previous name for the gene.

Authors should mention that SYP-4::HA staining does not necessarily imply that the tripartite SC is formed.

In *C. elegans* SYP-4 is required for assembly of the SC central region and localizes only to assembled SCs or polycomplexes. Although it is beyond the scope of this manuscript, we have found the same to be true for *P. pacificus.*

Revise use of italics throughout the text : ie Line 11, 13, 87.

These have been corrected.

J4- Please define this stage for non-experts.

We have now clarified that this is the final larval stage for *P. pristionchus.*

Define COSA-1 when first used.

We have now spelled out the full gene names for COSA-1 and CNTD1.

Class I and Class II crossovers are first defined in the Discussion section, but they are repeatedly referred to in the introduction and results. An earlier definition seems appropriate.

Agreed, but largely moot at this point since we no longer invoke the presence of Class II COs in *P. pacificus.*

For all genetic modifications (null mutations and tag insertion), indicate the protein change (truncation, deletion, position of tag…). Indicate whether epitope-tagged proteins are heterozygous or homozygous.Respect order of figure presentation: figure 5 cannot come in L178.

We felt that it was more logical to group this image with other examples of oocytes at diakinesis. If the editors feel strongly about this we can rearrange the order of the figures.

[Editors' note: further revisions were suggested prior to acceptance, as described below.]

Essential Revisions:1) Ideally to validate the DMC1 staining, the authors would need to exclude the possibility that it is not due to the V5 tag. The fact that RAD51::V5 behaves as RAD51 does not allow to conclude on the potential effect of the V5 tag on DMC1. Otherwise, the authors should mention explicitly the possibility that the broad DMC1 staining observed could be a specificity of the tagged protein.

In light of the very obvious differences between the distribution of RAD-51::V5 and DMC-1::V5, we felt that it was clear that neither the use of the V5 tag or the antibody used to detect it could have produced the much broader distribution we observed for DMC-1::V5. Nevertheless, we have now made an additional DMC-1::3xFLAG allele. In contrast to the DMC-1::V5 allele, the 3xFLAG-tagged allele was not fully functional: homozygous animals display a loss-of-function phenotype, but heterozygotes show normal meiotic progression, and the localization of DMC1::3xFLAG in these animals is indistinguishable from what we documented for DMC-1::V5. This further indicates that the broad distribution of DMC-1 relative to RAD-51 is not a consequence of either the tag or antibody used for immunolocalization. See Figure 4—figure supplement 2 (scale bar, 30 µm):

I will note that this additional validation required several weeks to complete, including several full days of hands-on time to design and inject new repair templates, screen many progeny to find insertions, verify them by sequencing, and to characterize and image a new allele, at a cost of several thousand dollars in materials and labor, as well as time that could have been used for other experiments. I mention this to encourage referees to carefully consider whether the effort and cost of “essential” revisions they request justifies the added value that they are likely to provide.

2) Data about orthologs and protein alignment should be provided (SPO11, DMC1, HOP2, MND1, RAD51, HOP1, SYP4, COSA1).

We are now including as Supplemental Data the sequences of homologous proteins from major model organisms and humans, along with multiple sequence alignments.

3) RAD-51 foci in wild type and Dmc1 mutant should be quantified to determine whether there is a detectable and significant difference.

In *dmc-1* mutants, RAD-51 foci are detected only after the very extended transition zone, probably only once the crossover assurance checkpoint is overridden (based on our knowledge of meiotic progression in *C. elegans*). These nuclei have a very different morphology and cell cycle status relative to the pachytene nuclei in which RAD-51 is normally detected in wild-type animals, and the chromosomes are unpaired and unsynapsed, which would be expected to impact the dynamics of DSB repair. Thus, we feel that this analysis would not provide a useful comparison to RAD-51 foci in wild-type animals.

L41. Is.

Corrected.

L122. Please explain this apparently synchronyzed group of cells in S phase.

We have now explained this in more detail (L122-124). Cy3-dUTP is delivered by microinjection and persists transiently in the gonad, and thus labels any nucleus that undergoes part or all of S-phase during this period. Some nuclei show preferential labeling of the X-chromosome, which is silenced in the germline and likely late-replicating. A similar pattern of incorporation is seen following injection of fluorescent nucleotides into the *C. elegans* germline (*cf*. https://doi.org/10.1016/j.ydbio.2007.05.019).

Lines 129-131, Figure 1A: the text sets up that the region of decondensation between diplotene and diakinesis has been called the "growth zone" in Ppa and the "diffuse stage" in other organisms, but the figure calls this region in Ppa "Diff" for diffuse stage. I am ok with this nomenclature, but the authors may want to say in the text or legend that for the sake of consistency with other organisms, they will call this the diffuse stage.

We have added this to the text.

L142. Figure 1D. Please clarify the staining in this panel D: one expects 12 DAPI staining bodies, or 12 CENP-C structures ? More than 12 are visible?

We have now clarified this in the text (L148). These are diploid mitotic nuclei, which have 12 DAPI-staining bodies/chromosomes. During mitosis each chromosome is comprised of two chromatids, each of which forms a kinetochore on opposing faces, so up to 24 linear CENP-C structures are observed.

L179. What do you mean by "frequent".

This has now been reworded/clarified in the text (L187-188).

L181. What is the plate phenotype?

We have reworded this for clarity (L189-190).

L189. Figures should be numbered following the text.

We have tried to present the figures in the same order they are called out.

Lines 200-216: For some reason, I got confused when I read, "in wild-type *C. elegans*" in line 212 and wasn't sure what genotype was described earlier. This was probably just due to a tired brain, but it might help the reader if line 204 began, "In zone 1 of wild-type *P. pacificus* hermaphrodites," and then "in wild-type animals" could be deleted from line 207. This way, the reader would have the genotype in their mind before seeing the data.

We thank the reviewer for the suggestion and have changed the text to describe pairing in the different zones and compare/contrast with *C. elegans* more clearly.

L218. Please clarify: in wild type, the pairing is decreased in the zones following meiotic entry, ie after the transition zone.

See above; we have reworded this description of pairing dynamics.

L220. Rad51 mutant seems to have a phenotype in zone 2?Please quantify heterologous distances in Rad51 mutant.

We have now included measurements of heterologous distances in Figure 1E. As to the differences between *rad-51* and wild type: we reexamined the source images that were quantified and found some genotype-independent variation in the length of the proliferative zones between wild-type and *rad-51* animals. These stochastic differences are often seen even between age-matched animals of the same genotype. Particularly because the region of fully paired nuclei is quite short in *P. pacificus*, such variation can result in minor variations in the representation of meiotic stages and thus interhomolog distances within each “zone” between datasets, but our evidence indicates that this small difference is not biologically significant.

Lines 274-287, Figure 3B: The nuclei labeled PM have puncta of SYP-4 and appear to have weak HOP-1 associated with chromosomes rather than excluded from chromosomes, which one might expect if the expression were truly nucleoplasmic. Is it possible that these nuclei are early leptotene, and that in P. pacificus adoption of the crescent shape is not concomitant with the onset of meiosis? It is curious that in the full gonad image in 3A, more distal nuclei do not appear to express SYP-4 or HOP-1, which makes me wonder if the "PM" nuclei are truly premeiotic or very early meiotic.

We thank the reviewer for pointing this out. Our pulse-labeling of replicating DNA (Figure 1B) shows incorporation in nuclei essentially up to the boundary where crescent-shaped nuclei appear, indicating that – as in *C. elegans* – crescent-shaped morphology occurs concomitantly with the onset of meiotic prophase. It does appear that HOP-1 and SYP-4, and likely other meiotic proteins, are expressed even prior to this stage, which is also the case in *C. elegans*. Whether we should consider the S-phase immediately prior to meiotic prophase to be part of meiosis has not really been established even in *C. elegans* and is beyond the scope of this work. We have now changed the text to state that the crescent-shaped morphology corresponds to the onset of meiotic prophase, rather than “meiosis” (L125-126).

L288. The authors should add a comment about the SYP-4 lines in Spo11-1 mutant (as indicated in the rebuttal letter)(non homologous? not the X?).

We have added comments to this effect (L327-329).

Line 305, etc: It's clearly beyond the scope of this manuscript, but it would be really interesting to look at DMC-1 localization in P. pacificus spo-11 mutants irradiated with a low dose of γ irradiation to see if at very low DSB number DMC-1 "coats" the chromosomes or forms puncta, and whether there is a DSB number or density at which staining switches to the broad distribution seen in wt.

We agree that such experiments could be interesting. However, little is known about how the amplitude of DNA damage signaling (*e.g*., ATM/ATR kinase activation) correlates with DSB numbers, even in *C. elegans*, and it would thus require extensive preliminary experiments to interpret the results of such an experiment. We are currently exploring the regulation of DMC-1 and RAD-51 through other approaches.

L338. Not necessarily dsDNA, DMC1 could interact with proteins.

We thank the reviewer for pointing this out and have now changed the wording in the text (L376).

L351: It seems that two alternatives are: two rounds of DSBs? or two rounds of DSB repair?

We favor the simpler idea that there is a transition in the mode of DSB repair, as has been inferred in *C. elegans* and other organisms but cannot rule out the possibility that synapsis triggers a new round of DSBs.

L381. Please comment on why egg viability is reduced.

Please see L408-409 in the revised text.

L468. How are the two arms defined?

They are defined cytologically. In both *C. elegans* and *P. pacificus*, crossovers are enriched on the “arms” of each bivalent, while the central region is relatively cool for crossing-over. The coupling of SC disassembly to CO site designation leads to the formation of a “long arm,” which retains cohesion during MI, and a “short arm” which retains SC through diakinesis, but then releases cohesion during MI. It is also possible that the short arm appears shorter in part due to a greater degree of compaction than the long arm, but this has not been assessed systematically.

L484 and 585. Why ? I understood the low CO activity is due to an inversion or other rearrangement, why then proposing a PC effect?

We thank the reviewer for this question, which highlights a gap in our explanation; we have now explained this important issue more clearly (L627-639). The absence of COs along entire chromosome arms could be the consequence of large inversions that suppress pairing/synapsis along the entire regions. However, it is also consistent with more limited structural heterozygosity together with the presence of a PC-like activity that promotes COs in a directional fashion from one side of the chromosome. We favor the latter explanation, in part because large inversions have not been detected between different strains of the same species in *Caenorhabditis* species or other nematodes. A complete genome assembly of each strain will help to resolve this issue but is beyond the scope of this work.

L536. This is an over-simplification, the mouse data is based on X ray irradiations.

We thank the reviewer for pointing out this caveat and have softened the language in mentioning this work.

Line 541: it is interesting that *C. elegans* rad-51 mutants have fragmented and "clumpy" chromosomes, and these are not seen in either Ppa dmc-1 or rad-51 mutants. Have you looked at a double mutant? I know this would be a tricky experiment without balancers (unless dmc-1 and rad-51 happen to be on the same chromosome).

They are not on the same chromosome, and we have not attempted this experiment, but it would indeed be interesting to look at this.

L991. This sentence does not belong to panel A.

Thank you for catching this – the misplaced title is a consequence of adding panel A to the figure. We have removed it.